# All-at-once RNA folding with 3D motif prediction framed by evolutionary information

Aayush Karan [1,2] & Elena Rivas [1,2] ✉

Structural RNAs exhibit a vast array of recurrent short three-dimensional (3D) elements found in loop regions involving non-Watson−Crick interactions that help arrange canonical double helices into tertiary structures. Here we present CaCoFold-R3D, a probabilistic grammar that predicts these RNA 3D motifs (also termed modules) jointly with RNA secondary structure over a sequence or alignment. CaCoFold-R3D uses evolutionary information present in an RNA alignment to reliably identify canonical helices (including pseudoknots) by covariation. Here we further introduce the R3D grammars, which also exploit helix covariation that constrains the positioning of the mostly noncovarying RNA 3D motifs. Our method runs predictions over an almost-exhaustive list of over 50 known RNA motifs ('everything'). Motifs can appear in any nonhelical loop region (including three-way, four-way and higher junctions) ('everywhere'). All structural motifs as well as the canonical helices are arranged into one single structure predicted by one single joint probabilistic grammar ('all-at-once'). Our results demonstrate that CaCoFold-R3D is a valid alternative for predicting the all-residue interactions present in a RNA 3D structure. CaCoFold-R3D is fast and easily customizable for novel motif discovery and shows promising value both as a strong input for deep learning approaches to all-atom structure prediction as well as toward guiding RNA design as drug targets for therapeutic small molecules.

Many noncoding RNAs play essential roles in cellular processes by means of conserved three-dimensional (3D) structures[1]. Accurately determining the 3D structure of an RNA is a window into inferring its molecular mechanism of action.

RNA structure is hierarchical. Canonical base pairs (*cis*-Watson−Crick A:U, G:C and G:U wobble pairs) stack together as double helices and pseudoknots, forming the secondary structure. Critical loops and junctions connect these helices and arrange them into a 3D structure. These nonhelical linker regions, called RNA 3D motifs[2] or modules[3], have been extensively studied in the literature[4–24] for their importance

in accurately characterizing full RNA structure. The RNA 3D motifs have a few general properties: they are typically short; they include recurrent patterns of non-Watson−Crick base pairs resulting in complex and distinctive 3D architectures; and often, they also display conserved sequence patterns. Their structural properties are usually independent of the helices they are connected to, and thus, identifying 3D motifs alongside secondary structure provides important additive clues that guide the assembly of a full RNA structure from its sequence.

RNA 3D motifs (modules) are inherently difficult to detect due to their short size (often between 4 and 20 nucleotides), sequence

[1]Department of Molecular and Cellular Biology, Harvard University, Cambridge, MA, USA. [2]These authors contributed equally: Aayush Karan, Elena Rivas. ✉e-mail: elenarivas@fas.harvard.edu

variability within motif types and their sheer variety (more than 30 well-categorized motifs have been identified in RNA crystal structures[11] from the Protein Data Bank[25]). They can also be discontinuous in linear sequence, and they can appear in internal loops or junctions where the fragments composing the motif are hundreds of nucleotides apart. Important efforts have been developed to extract RNA 3D motifs from crystal structures and to create databases of RNA 3D motifs, such as: RAG[26,27], FR3D Motif library[28], RNA3DMotif[19], RNA FRABASE[29], RNA 3D Motif Atlas[30], RNA Bricks[31], CaRNAval[32], LORA[33], D-ORB[34] and ARTEM[35]. Based on this knowledge, several important efforts exist to predict RNA 3D motifs from sequence such as RMDetect[3], JAR3D[36], RMfam[37] and BayesPairing2[38].

However, these methods are not fully integrated with secondary structure prediction. Several methods[3,36,38,39] are indirectly guided by secondary structures predicted by standard thermodynamic methods[40,41]. However, because those thermodynamic methods cannot incorporate similar parameters for the 3D motifs, the prediction of motifs cannot be integrated together with that of canonical base pairs. In fact, the inputs required can be quite strong: for example,[36] requires that the loop regions testing for the presence of motifs are provided, whereas[38] trains over annotated motifs in one family for prediction, getting the most competitive results only when the train and test family are the same. Furthermore, previous techniques[3,36,37] are computationally expensive, making independent predictions for one motif at a time. This also restricts the diversity of motifs predicted over, often relegated to hairpin and internal loop motifs[3,36].

Here, we introduce CaCoFold-R3D, a computationally fast probabilistic model that simultaneously predicts the joint RNA 3D motifs and secondary structure present in a structural RNA. CaCoFold-R3D is grounded on the power of covariation in alignments as inputs. Although covariation is not prominent in RNA 3D motifs, the covariation found in canonical helices constraints the space where these 3D motifs can occur, and R-scape's covariation analysis[42] assigns statistical significance as to whether its predictions are evolutionarily conserved RNA structures[43]. Methods such as RMDetect[3] and BayesPairing2[38] also use alignments, but they do not provide statistical significance for their predictions.

Another important feature of CaCoFold-R3D is the exclusive use of probabilistic modeling, which naturally facilitates the integration of the prediction of RNA 3D motifs with that of the RNA secondary structure. Several existing methods use probabilistic modeling of RNA 3D motifs, but they do not integrate those with the predictions of canonical base pairs[3,36,37]. CaCoFold-R3D deploys an array of stochastic context-free grammars (SCFGs) to model the structural architecture, and profile hidden Markov models (HMMs) to model sequence homology, incorporating a large variety of motifs—accounting for sequence variability, we predict over 96 motifs total present in any loop region including hairpins, bulges, internal loops and multiloops. In addition, the CaCoFold-R3D grammar is designed to generate not just individual sequences but probabilistic sequences representing the columns of an alignment. This important feature allows the modeling of sequence variations within the motif.

CaCoFold-R3D serves as a structural paradigm for a new class of probabilistic RNA folding algorithms that directly integrates the prediction of multiple RNA 3D motifs with that of canonical helices, as well as triplets and other long-range interactions, all of that constrained by the covariation found in the input alignments.

## Results

### CaCoFold-R3D: covariation-contrained RNA 3D motif prediction

Figure 1 describes the overall CaCoFold-R3D method. The input is a sequence or alignment, and the output is an RNA structure that includes RNA 3D motifs and canonical helices (both nested and pseudoknotted), as well as other tertiary base pairing interactions, provided that they have covariation evidence.

From an alignment, R-scape identifies a set of positive base pairs that significantly covary above phylogenetic expectation and a set of negative pairs that are not expected to form because their variability is not reflective of them being base paired[42,44]. We have previously shown that the accuracy of RNA structure prediction improves significantly by using covariation information as prediction constraints[45]. Crucially though, CaCoFold-R3D not only uses covariation to constrain secondary structure prediction, but it further uses covariation-bound secondary structure to further constrain the location of RNA 3D motifs via an integrated SCFG.

Specifically, CaCoFold-R3D splits the covarying pairs into layers each with the maximum number of nested pairs until all positive pairs have been taken into account. The first layer includes the maximal number of covarying nested base pairs and is folded into the main secondary structure. The rest of the layers are expected to identify helices of pseudoknotted canonical helices and other tertiary base pair interactions provided that they have covariation support. CaCoFold-R3D introduces a novel SCFG called RGBJ3J4-R3D to describe the first layer where the main structure is predicted. RGBJ3J4-R3D jointly infers the collection of nested canonical helices along with the RNA 3D motifs found within the loop regions (Fig. 1) via a maximum probability parsing facilitated by dynamic programming.

### Comparison with other methods

Our method builds on several previous methods to identify RNA structural elements beyond Watson–Crick base pairs forming a nested secondary structure.

Some methods attempt to extend the repertoire of interactions beyond canonical Watson–Crick base pairs. Methods such as MC-Fold[46] and RNA-MoIP[47] (which includes MC-Fold) build a secondary structure with a model that introduces special pseudo-energy scores for some small whole hairpin loops and internal loops. However, the output does not identify any structural 3D motifs per se, and for example, it would not be able to distinguish a K-turn from any other 2 × 5 internal loop. Another method that includes non-Watson–Crick interactions is the method RNAwolf[48], which also introduces terms to predict some triplets. However, RNAwolf cannot tell whether those triplets belong or constitute any of the known 3D motifs. CaCoFold-R3D directly predicts 3D motifs, thus a direct comparison is not possible.

Methods that try to identify specific 3D motifs include JAR3D[36] and RMfam[37]. These methods require being given the loops in which to look for target 3D motifs, and the search is done for a small number of motifs and one at the time.

The closest methods to CaCoFold-3D are RMDetect[3] and Bayes-Pairing2[38]. These methods model specific 3D motifs, but the number of motifs predicted is limited, and the motifs are identified one at a time. Both methods require the previous specification or prediction of a secondary structure. The methods are also computationally expensive. BayesPairing2, the fastest of the two approaches, has to process RNAs longer than 300 nucleotides by windowing. CaCoFold-R3D on the other hand integrates the prediction of the 3D motifs with that of the secondary structure, and it can deal with even the largest ribosomal RNA molecules. With respect to describing the 3D motifs, RMDetect uses Bayesian Networks which precisely describe all non-Watson–Crick base pairs in the motif. This approach, however, is inherently computationally expensive. BayesPairing2 improves time performance by using much faster regular expressions to preselect loops in which to apply the Bayesian Networks. Instead, CaCoFold-R3D uses SCFGs with profile HMMs to model the motifs. These SCFGs lack the detail of the Bayesian networks in describing every single base pair but still describe correlations between loops effectively, and importantly, they can be integrated into standard dynamical programming algorithms for RNA secondary structure prediction such as CaCoFold[45]. Thus CaCoFold-R3D has the same time complexity as standard methods for just secondary structure prediction.

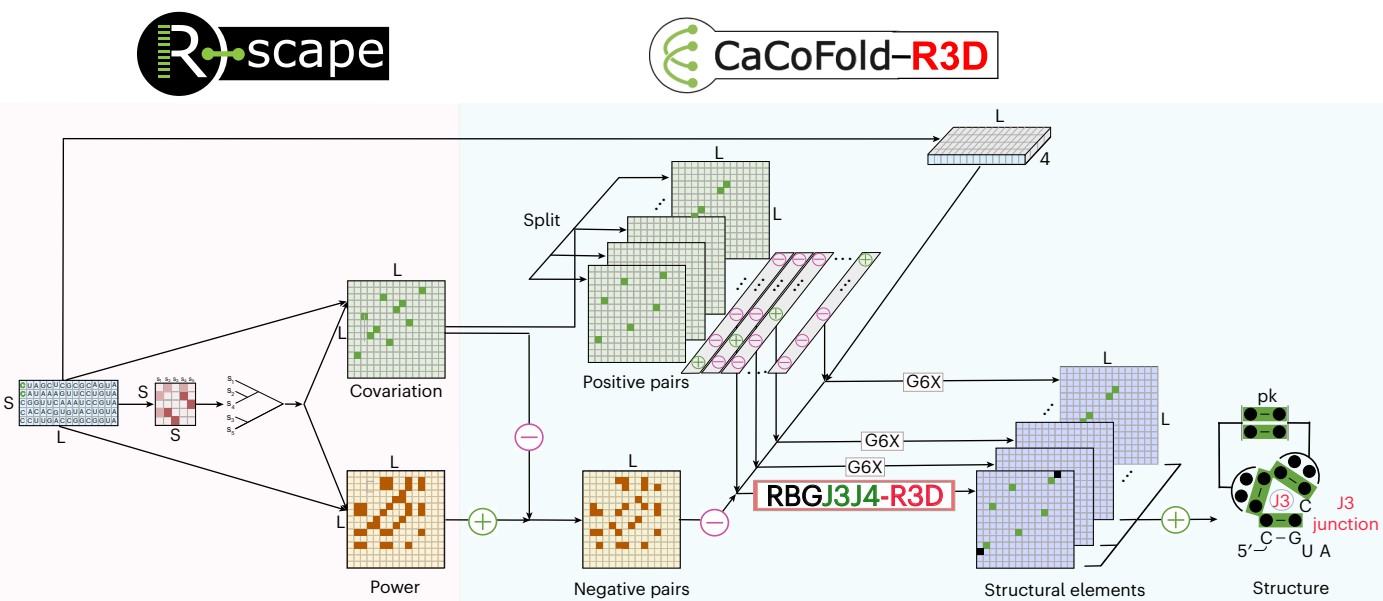

**Fig. 1 | The CaCoFold-R3D algorithm for the joint prediction of 3D RNA structural motifs integrated with canonical RNA helices.** The end-to-end method using a toy alignment of length $L = 15$ and $S = 5$ sequence is shown. The R-scape identifies significantly covarying (above phylogenetic expectation) base pairs and the positive pairs, as well as negative pairs that have evidence against being base paired. CaCoFold-R3D produces a structure that includes canonical helices, as well as 3D motifs using a layered approach that uses different probabilistic RNA folding grammars. The first layer includes the largest set of positive pairs that are nested with each other and uses the RGBJ3J4-R3D grammar to predict a secondary structure including 3D motifs. The rest of the layers use the G6X grammar[44], which adds pseudoknots and other tertiary base pair interactions with covariation evidence. The toy alignment has 5 significantly covarying base pairs (green) and 13 negative base pairs. CaCoFold-R3D needs to use two layers, and the resulting structure includes three nested helices, another helix forming a pseudoknot and has one annotated three-way junction 'J3'. The black dots in the consensus structure indicate that there is not a consensus nucleotide for that position.

CaCoFold-R3D has a collection of uniquely desirable properties not found together in any previous method: (1) the method can handle any 3D motif occurring in hairpin loops, bulges, internal loops, three-way (J3) and four-way (J4) junctions, as well as independent sequence motifs in branches of higher-order junctions. (2) All motifs are predicted at once and under one unique probabilistic model. (3) The model can folds entire alignments, taking into account RNA 3D motif sequence variability even within a given structural RNA family. (4) The method is computationally efficient. (5) It has the potential to facilitate novel motif discovery. (6) Importantly, CaCoFold-R3D uses the evolutionary signal found in the alignment to inform the prediction both of canonical helices and 3D motifs which significantly improves performance and provides statistical confidence on the predictions depending on the amount of covariation observed in the helices bounding the motif.

### RBGJ3J4-R3D: one single SCFG predicts helices and 3D motifs

The RGBJ3J4-R3D model described in Fig. 2 is an SCFG that simultaneously infers the secondary structure of nested canonical helices as well as the RNA 3D motifs present in any of the loop regions. It combines together a grammar called RBGJ3J4 (Methods and Extended Data Fig. 1), with a library called R3D of RNA 3D motif grammars described in the next section. RBGJ3J4 is unique in that it has specific descriptions for J3 and J4 junctions that are the most frequent of the multiloop structures found in RNA structures, which form many different RNA 3D motifs present in important RNA molecules such as the hammerhead J3[49] and the J4 of the hepatitis C virus IRES[50].

RGBJ3J4-R3D creates specific R3D grammar models (that is, grammar nonterminals) for each of the different loop motifs. To incorporate these motif nonterminals into the RBGJ3J4 grammar, we simply add the motif SCFGs as additional productions along with a generic loop motif (Fig. 2). Motif designs are added for six classes of loops: hairpin (HL), bulge (BL) and internal loops (IL), as well as J3 and J4

junctions, and general branch segments (BS) that can appear in any branch of any higher-order multiloop.

**Training of the RBGJ3J4 probabilistic parameters.** The RBGJ3J4 grammar (Fig. 2a) depends on parameters that describe the probabilities of the different rules permitted by a given nonterminal, as well as emission probabilities to describe single unpaired or in loops residues, as well as paired and stacked paired residues. As the RNA basic grammar (RBG) grammar in CaCoFold[45], the RBGJ3J4 grammar has been trained by maximum likelihood using TORNADO[51] on a large and diverse dataset of known RNA structures and sequences. This dataset (TrainSetA + 2 × TrainSetB) introduced in TORNADO includes a total of over 4,000 different sequence/structure pairs, spanning over 100 different RNA structures (see Fig. 2 in TORNADO[51] for more details). The large diversity of the training set guarantees that SCFGs trained on it do not overfit (on a TestSetA+TestSetB also introduced in TORNADO) relative to thermodynamic models not trained on sequence data (Fig. 5 in TORNADO[51]).

The main difference between the RBG and RBGJ3J4 grammars is in the multiloop (ML) nonterminal (Fig. 1). The RBG single rule for ML splits into three rules for RBGJ3J4 representing the relative frequencies of J3, J4 versus other higher-order junctions. For the training set above, 51% of all multiloops are J3 junctions, and 32% are J4 junctions, which stresses the importance of these lower-order junctions amongst all possible multiloops. The collection of all probabilistic parameters for the RBGJ3J4 and RBG SCFGs are provided as part of the code in file covgrammars.c.

**Parameterization of RBGJ3J4-R3D probabilistic parameters.** The RBGJ3J4-R3D grammar modifies the nine rules in RBGJ3J4 that are responsible for the incorporation of loops into the structure (Fig. 2b, red arrows). These rules describing generic loops are expanded into also describing specific 3D motifs in that class. The additional transition probability parameters associated to those new 3D motif-specific

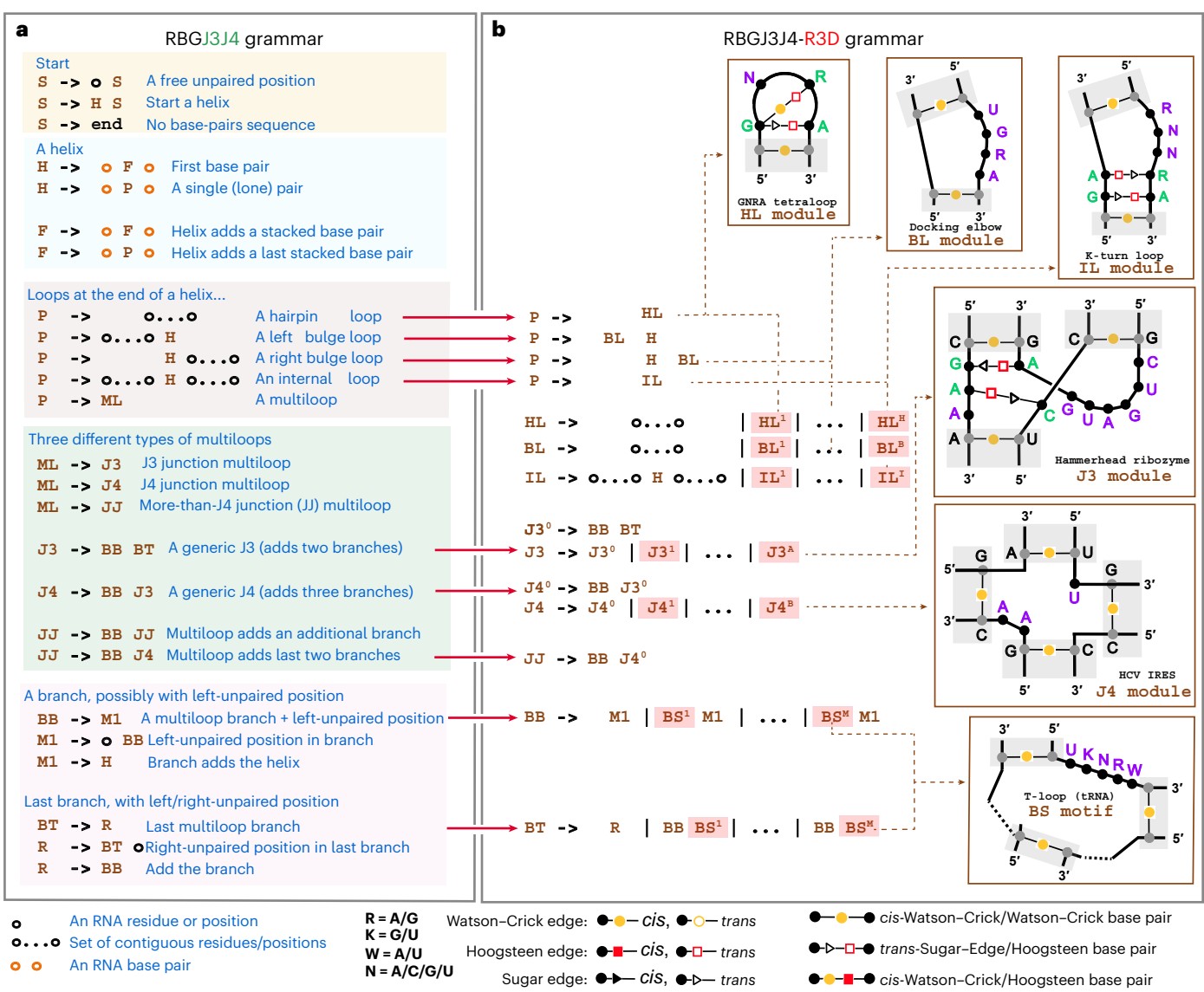

**Fig. 2 | The RBGJ3J4-R3D grammar. a,b**, The extended RBGJ3J4-R3D grammar with the modified elements describing specific RNA 3D motifs is highlighted in red. Individual examples of RNA structural motifs and their corresponding appearance in the model are given as inserts. We depict in gray Watson–Crick base paired residues in the canonical helices setting the bounds but not being part of the RNA motif. In green, the positions of the motif involved in non-Watson–Crick base pairing are shown, and in purple, the motif positions not paired are shown. The RBGJ3J4-R3D grammar can be used to describe alignments not just individual sequences. The BB/M1 non-terminals (as well as the BT/R) are redundant for the RBGJ3J4 grammar in **a** (see supplemental Figure S1 for a simpled description), but they become different entities in the RBGJ3J4-R3D grammar in **b**. The RBGJ3J4-R3D grammar is unambiguous, that is, a given alignment with a particular arrangement of base pairs and 3D motifs can only be generated one way by the grammar.

rules could in principle be calculated by maximum likelihood as the rest of the parameters before in RBGJ3J4. Unfortunately, there is not yet a reliable and large enough database of RNA sequences/structures that extensively annotates 3D motifs along with the secondary structure.

While such database of RNA secondary structures and 3D motifs is produced, we have proceeded as follows: the RBGJ3J4 probability of a given loop type is split between the generic loop state and the whole R3D motif class, which gets assigned a fraction of it. Those fractions, set for now by human curation, are (0.4, 0.4, 0.5, 0.2, 0.2, 0, 2) for the hairpin loop, bulge loop, internal loop, J3, J4 and branch segments motif types respectively. Moreover—and to avoid any possible overfitting—we use the principle of maximum entropy and assign the same probability to each of the specific 3D motifs of a particular class given in the descriptor. For instance, the probability of forming a generic hairpin loop in the trained RBGJ3J4 is 0.3475, thus RBGJ3J4-R3D assigns

0.2085 (= 0.3475 × 0.6) to the generic hairpin loop, and the remaining 0.1390 (= 0.3475 × 0.4) is equally distributed over all defined hairpin loop motifs (15 in the current implementation). Thus, each 3D hairpin loop motif gets assigned a probability of 0.0093 (= 0.1390/15).

Next we describe the specific R3D models for all six different loop classes. The R3D grammars incorporate an arbitrary number of 3D motifs in any arbitrary loop region into the folding grammar. Integrating the R3D grammars with the RBGJ3J4 grammar gives one SCFG jointly modeling both secondary structure and motifs (Fig. 2).

## R3D: six architectures describe motifs in all types of loops

The key insight behind the R3D grammars is to realize that RNA 3D motifs have a structural component determined by the set of (mostly conserved) non-Watson–Crick pairs that characterize the motif and also a sequence-based component as many 3D motifs also conserve

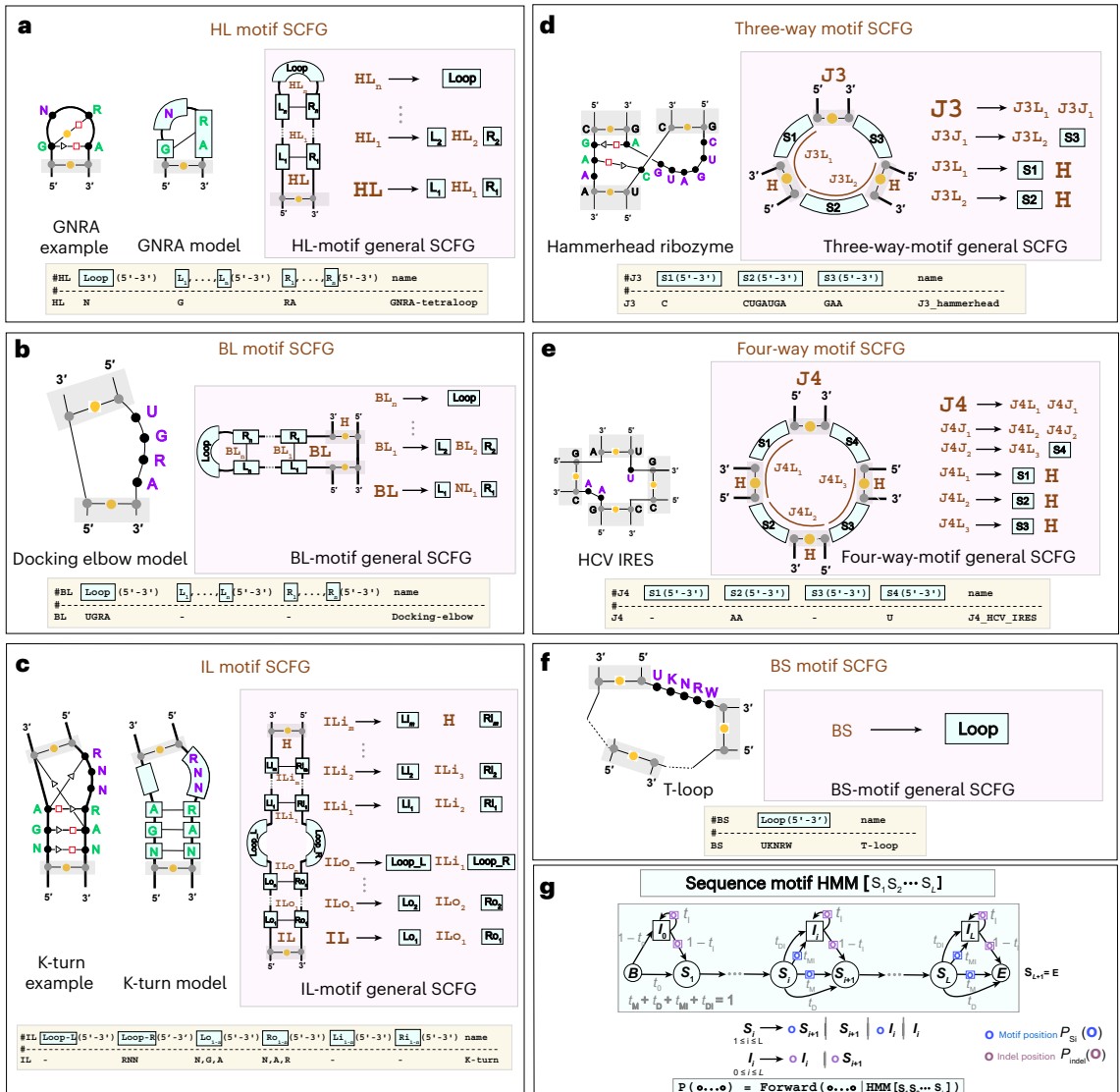

**Fig. 3 | The R3D grammars describe six types of RNA 3D motif. a–f**, The motif types: hairpin loop (HL) (**a**), bulge loop (BL) (**b**), internal loop (IL) (**c**), J3 junction (**d**), J4 junction (**e**) and branch segment (BS) (**f**). For each motif, the bold terminal: HL, BL, IL, J3, J4 and BS, is the one that inserts into the RBGJ3J4-R3D grammar in Fig. 2. Closing Watson–Crick base pairs are not included in the motif models. In yellow, the descriptors used to build any of the six motifs. **g**, The sequence string modeled with profile HMMs (blue) which uses a start non-terminal B, and sequence non-terminals $S_i$ and $I_i$ for a match and insert at position $i$, respectively.

particular residue identities. The R3D grammars describe the structural component of a motif using profile SCFGs specific for each type of motif (Fig. 3a–f) and the sequence component with customized profile HMMs that permit sequence variability (Fig. 3g).

The key that makes the R3D grammar affordable is that unlike other methods like RMDetect[3] or BayesPairing2[38], R3D does not attempt to model each of the actual non-Watson–Crick base pairs individually (which can be quite complicated and non-nested). R3D instead models groups of residues that are correlated because of their underlying non-Watson–Crick base pairing. This induces a segmentation of a motif into continuous subsequences (modeled by profile HMMs) involved in specific correlations (modeled by the SCFGs). This decomposition allows one model to describe all motifs of one given type, giving rise to a generalized R3D grammar per motif type (Fig. 3). The SCFG states can generate multiresidue long strings using specific profile HMMs. For each motif, the individual nucleotide bases that constitute each segment of the profile motif are of course dependent on the consensus sequences of the motifs.

We consider six different types of structural motif, based on whether they occur in hairpin, bulge or internal loops, as well as in J3,

J4 or branch segments that can occur in any junction. Each of the six general R3D SCFG models in Fig. 3a–f have a particular SCFG architecture describing the interactions present in each motif.

We decompose a motif into helical and nonhelical interactions. Helical interactions are modeled by classic Watson–Crick pairing. The nonhelical interactions constitute the actual motif and there are broken down into contiguous subsequences (sequence segments) that belong to loops; for example, just one segment in one loop capping a helix corresponds to a hairpin loop motif, while three separate loops connecting three helices correspond to a J3 junction. In the Methods, we detail the segmentation method per type of motif, along with the corresponding grammar rules.

**Parameterization of R3D profile HMMs.** The 3D motifs behave mostly independently of the RNA structure in which they appear. This adaptability has facilitated the identification of 3D motifs repeatedly found in the context of very different RNA structure[11], and now it facilitates the parameterization of the models describing these motifs in a way that naturally generalizes to any RNA structures in where they appear.

In this work, each individual motif is assigned a R3D profile SCFG that belongs to one of the six classes in Fig. 3a–f. Each motif includes a particular set of profile HMMs describing the actual sequence motifs defining the motif. For instance, any HL motif has one 'loop', and at least two correlated L-R profiles (Fig. 3a). Each profile HMM has a consensus sequence. As an example, we design the GNRA tetraloop (an HL motif) with a one-nucleotide profile HMM loop = 'N', an two correlated profiles L = 'G' and R = 'RA', as 'G' interacts both with 'R' and 'A' (Extended Data Fig. 3). Profile HMMs can have an empty consensus. For instance, the U-turn three-loop motif has a loop = 'URA' profile, but both the L and R profiles are empty.

A profile HMM assigns one match state S per consensus position (Fig. 3g). Each S state assigns the same probability to all residues in the consensus for that position, and mismatches to residues not in the consensus are allowed with a small probability of $10^{-4}$. For instance, for the U-turn motif profile HMM loop = 'URA', the emission probabilities for the first position 'U' are $p(U) = 0.9997$ and $p(A) = p(C) = p(G) = 0.0001$. The emission probabilities for the second position 'R' are $p(A) = p(G) = 0.4999$ and $p(C) = p(U) = 0.0001$. With this parameterization, CaCoFold-R3D is able to identify for instance a U-turn motif 'YAA' in the Metazoan SRP.

For each consensus S state, there is also an insertion state I, and a deletion path which together allow to extend or shorten the size of a given motif to model the variability observed for the actual motifs found in the 3D structures[25]. In this implementation, the transition probabilities between states are parameterized such that the profile HMM generates sequences that on average slightly exceed the length of the consensus motif (adding a 0.1 per position) up to a max of 1.5 extra length per motif on average. With this parameterization, while we have designed the loop-E IL motif to have five consensus positions on each branch, CaCoFold-R3D is able to identify the loop-E in 5S rRNA, with five and six residues per branch, respectively.

Another source of variability occurs because alignments can introduce large insertions in loop regions that extend beyond the actual 3D motif. By default, R-scape does not analyze positions in the alignment that have more than 75% gaps (by default). This effectively removes highly variable regions from analysis, which helps identifying the actual 3D motifs that generally show a high level of conservation.

**Motif variants.** All RNA 3D motifs except hairpin loop (HL) motifs are bound by more than one helix, thus allowing different topological variants depending on which 5'/3' ends are selected to integrate the motif into the rest of the structure. Bulge and internal loop motifs have two variants, and J3 and J4 junctions have three and four variants, respectively. For instance, the two variants of any bulge loop motif correspond to a left and right bulge motif, respectively. Extended Data Fig. 2 describes all motif variants with their SCFG rules. For any 3D motif entry in the R3D descriptor file, CaCoFold-R3D internally models all possible variants of the motif.

**R3D-prototype: the importance of framing motifs by evolution**
One of the keys to our approach is that the CaCoFold-R3D method bounds the search of RNA 3D motifs to the segments of the RNA molecule enclosed by helical regions with covariation support. This is important, as, due to the small size of the motifs, their associated models have low information content and would otherwise produce large number of false positives.

To initially test the effect of adding covariation information into the prediction of RNA 3D motifs, we implemented a R3D-prototype that simultaneously produce a secondary structure and models two 3D motifs: the GNRA tetraloop (a hairpin motif) and the K-turn (an internal loop motif). This prototype uses a version of the RBG grammar (Extended Data Fig. 1a) that produces structural predictions directly on RNA sequences and implements two R3D grammars (also on sequences) modeling GNRA loops and K-turns. The prototype uses this RBG-R3D grammar and, for each RNA sequence, predicts a maximum probability secondary structure including GNRA loops and K-turn motifs. The R3D-prototype is independent of R-scape and does not assess significant covariation. But as input to the prototype one can add, as an option, external information of covarying base pairs.

For each Rfam family, sequences are selected at random from their seed alignments, and covarying base pairs are extracted from the Rfam seed alignments. We record both the sensitivity, defined to be the percent of truth motifs successfully detected, as well as the average number of false positives per prediction. We perform this analysis both including and excluding covariation information to demonstrate the effectiveness of the model.

In Table 1, we present results from applying the R3D-prototype to structural RNAs from different Rfam families[52]. As positives, we tested the U3 small nuclear RNA and the spliceosomal U4 RNA, which include two and one K-turns, respectively[53], and the 5S rRNA, which contains a GNRA tetraloop[54]. The U3 and U4 RNAs also serve as negative tests for the GNRA tetraloop and 5S rRNA as negative for the K-turn. For an independent control, we selected the 6S RNA and the ribosome modulation factor (RMF) RNA, which lack either of the tested motifs.

As hypothesized, adding covariation information vastly improves motif prediction accuracy despite the lack of covariation within the motifs themselves. Overall sensitivity on the detection of GNRA tetraloops and K-turns in the three positive RNAs increases after adding covariation from 84% to 95% (Table 1). Adding covariation also significantly reduces false positives for K-turn detection to similar levels to that of the GNRA tetraloop.

To further test the efficacy of our method, we applied it to four K-turns recently identified in bacterial RNAs via structural prediction and X-ray crystallography[55]. The performance of our method on these alignments corroborates the high level of accuracy and low false positivity as demonstrated before (Table 1). This R3D-prototype shows that our approach is a reliable predictor of confirmed motif structure. We moved on to making a full implementation of the RBGJ3J4-R3D model, named CaCoFold-R3D, that incorporates a large collection of RNA 3D motifs found recurrently in RNA structures[10,11,18] and operates on alignments.

**R3D SCFG profiles of over 50 recurrent RNA 3D motifs**
For the presented version of CaCoFold-R3D, we wanted to test its capability to integrate a comprehensive collection of RNA 3D motifs. We searched the literature for methods that had created 3D motif models, from which we finally cataloged 51 different motif architectures with confirmed representation in multiple structured RNAs[3,11,18,19,21,26,28–33,37]. It should also be noted that these does not have to be considered a definite list. The selection of motifs in CaCoFold-R3D is extremely flexible, and it only requires to adjust or modify the descriptor file. A customized descriptor file can be provided using the option --r3dfile <descriptor-file>. The R3D descriptor describing the 51 motifs is provided in Extended Data Fig. 3. For this particular R3D descriptor, the three variants for the J3 motif J3_typeB are redundant with each other, as well as the two variants of the internal loop motif J4/5-internal loop, and the total number of unique motif variants is 96. The R3D code, given a descriptor file, calculates automatically all possible variants and eliminates those that are redundant. The total number of total nonredundant RNA motifs used is reported as part of the standard output for further inspection.

Figure 4 includes a representation of 20 (out of 51) motifs included in this implementation. The full list of motifs can be found in Extended Data Table 1 and Extended Data Fig. 3, which also provides the descriptive notation used in our input files to represent the motifs in our models (Extended Data Fig. 2). The method is customizable by

**Table 1 | R3D-prototype accuracy on GNRA and K-turn motifs**

| RNA family | Number of sequences | Average length | Number of base pairs covary/total | Constrained by covariation | | | No covariation used | | |
|---|---|---|---|---|---|---|---|---|---|
| | | | | Sensitivity | False positives/sequence | | Sensitivity | False positives/sequence | |
| | | | | (%) | GNRA | K-turn | (%) | GNRA | K-turn |
| GNRA motif | | | | | | | | | |
| 5S rRNA (RF00001) | 50 | 117 | 32/34 | 100 | 0.14 | 0 | 80.65 | 0.29 | 0.24 |
| K-turn motif | | | | | | | | | |
| U3 snoRNA (RF00012) | 50 | 209 | 13/63 | 90.3 | 0.12 | 0.06 | 84.95 | 0.14 | 0.15 |
| U4 snRNA (RF00015) | 50 | 144 | 24/28 | 96.0 | 0.30 | 0.02 | 88.00 | 0.30 | 0.10 |
| Controls | | | | | | | | | |
| 6S RNA (RF00013) | 50 | 180 | 35/52 | – | 0.02 | 0.06 | – | 0.04 | 0.12 |
| RMF RNA (RF01755) | 50 | 130 | 1/28 | – | 0 | 0 | – | 0 | 0 |
| Random shuffle | 20 × 5 | (from all five | | – | 0.19 | 0.12 | – | 0.07 | 0.44 |
| Weighted total | | families) | | **95.4** | **0.15** | **0.06** | **84.5** | **0.14** | **0.24** |
| New K-turns | | | | | | | | | |
| Drum bacterial RNA (RF02958) | 100 | 113 | 22/29 | 93.0 | – | 0 | 89.5 | – | 0.09 |
| Actinomyces-1 RNA (RF02928) | 100 | 110 | 11/34 | 100 | – | 0.01 | 98.9 | – | 0.06 |
| RAGATH-18 RNA (RF03064) | 100 | 71 | 15/18 | 97.9 | – | 0 | 90.6 | – | 0.11 |
| HOLDH RNA (RF02997) | 10 | 401 | 6/95 | 100 | – | 0.64 | 80.0 | – | 0.60 |
| Weighted total | | | | **97.8** | – | **0.03** | **92.6** | – | **0.10** |

The tested RNAs include one of the two tested motifs: the GNRA tetraloop for 5S RNA sequences and the K-turn internal loop for the U3 and U4 snRNA sequences or none for the control 6S and RMF RNAs. Sensitivity measures the fraction of motifs are identified correctly by the R3D-prototype, where correct identification requires exact matching of the ends of the motif. False positives measure the instances of the tested motif found at the wrong position or instances of the not tested motif. For each RNA, sequences are selected at random from the corresponding alignments. The weighted averages are calculated weighting by the fraction of sequences of a particular RNA type that were analyzed. Several RNAs in the 'New K-turns' table also have GNRA loops, thus they cannot be used to estimate GNRA false positives. Bold values indicate best performance. The location in the alignments of the motifs and covarying base pairs are given in the Supplementary Information.

simply changing the input file with the representations of new motifs to be considered.

Figure 4 also includes for each of the 20 motifs a positive example of a Rfam family documented to have the motif, accompanied by a detail of the CaCoFold-R3D full structural prediction correctly detecting the motif. It is worth noticing, that in the majority of cases, the Rfam 3D motif is bounded by helices that show some level of covariation, further supporting to our key design feature of informing motif detection with the evolutionary conservation of secondary structure helices that arrange into a 3D structure.

**Results on RFAM alignments**

We ran CaCoFold-R3D on all Rfam seed alignments. CaCoFold-R3D finds the K-turns in the alignments of the U3 snoRNA[56,57], U4 snRNA[58] and the other four new K-turns[55] that we used in the R3D-prototype, as well as the K-turn in the SAM riboswitch[59]. Figure 5 reports additional examples of full structure predictions with representative 3D motifs that have been reported in the literature. For a comprehensive report of CaCoFold-R3D's performance on detecting known 3D motifs in their associated alignments, see Extended Data Table 1. Of the 44 motifs listed there, CaCoFold-R3D is able to detect all but three.

CaCoFold-R3D also identifies the GNRA tetraloop in the 5S rRNA tested with the R3D-prototype (Table 1). In the 5S rRNA[60], we also observe the Loop E motif, the two G-bulge motifs in the T-box riboswitch[61], the J4a/4b 3D motif of the Magnesium riboswitch[62] and the T-loop motif in the TPP riboswitch, as well as its characteristic J3 junction[63]. Two interesting cases are the CsrB RNA that binds to the CsrA protein[64], for which we identify 12 binding motifs, and the Glutamine riboswitch, where two R3D branch segment motifs allow us to identify a confirmed loop E motif occurring in a multiloop involving a pseudoknot instead of in an internal loop[65]. For the Metazoan SRP, CaCoFold-R3D identifies several of its characterized motifs (domain IV, C-loop, K-turn, U-turn and GNRAs)[66-68] (Fig. 5). In the Extended Data Table 2, we report the performance for all Rfam families. The collection of all Rfam predicted structures is provided in the Supplementary Information.

With regard to the distribution of detected 3D motifs, we observe that the GNRA tetraloop is the most frequently observed motif, followed by the K-turn. Most motifs of any other kind have between 10 and 50 instances in Rfam (Extended Data Table 1). Because the CaCoFold-R3D predictions integrate the covariation information observed in the alignment base pairs, we use the covariation observed in the helices bounding the 3D motifs to assess our confidence in the predictions. Overall, we detect a total of 2,124 motifs, of which 1,460 have covariation support, defined here as a motif for which at least one of its bounding helices has one or more covarying base pairs. A total of 591 of the Rfam families include 3D motifs with covariation support. For the two largest RNA structures the small subunit (SSU) and the large subunit (LSU) rRNA, we find 45 supported 3D motifs for the eukaryotic SSU and 62 for the eukaryotic LSU rRNA (Extended Data Table 1). The complete list of motifs detected for each Rfam family is also provided in the Supplementary Information.

As a control, we obtain predictions for negative alignments obtained from the Rfam alignment by permuting the residues in each column (position) independently from each other. As a result of the shuffling, the covariation signal in the input alignment is altered, but the base composition of the positions remain unchanged, thus retaining the sequence signature of any potential motif. For these control alignments, we obtain 121 motifs supported by covariation, which compared with the 1,460 motifs obtained for the Rfam alignments, indicate an estimated 8.3% false discovery rate in our predictions supported by covariation. For motifs not supported by covariation support, the comparison between found in Rfam (2,124 − 1,460 = 664) versus those found in controls (290 − 121 = 169) report an false discovery rate increases of 25.4%. Again, stressing the important effect of using covariation information in the detection of 3D motifs.

Notice that the control alignments report 733 helices out of 14,146 with at least one covarying base pair. Since R-scape[42,45] reports pair with a significance $E$-value cutoff of 0.05, this number (733) is in good agreement with the expected average number of helices with at least one covarying pair under the null hypothesis (707.3 = 0.05 × 14,146).

The Rfam alignments are good-quality structural alignments built with the method Infernal[69] using diverse homologs of known structural RNAs. When building RNA alignments several issues need

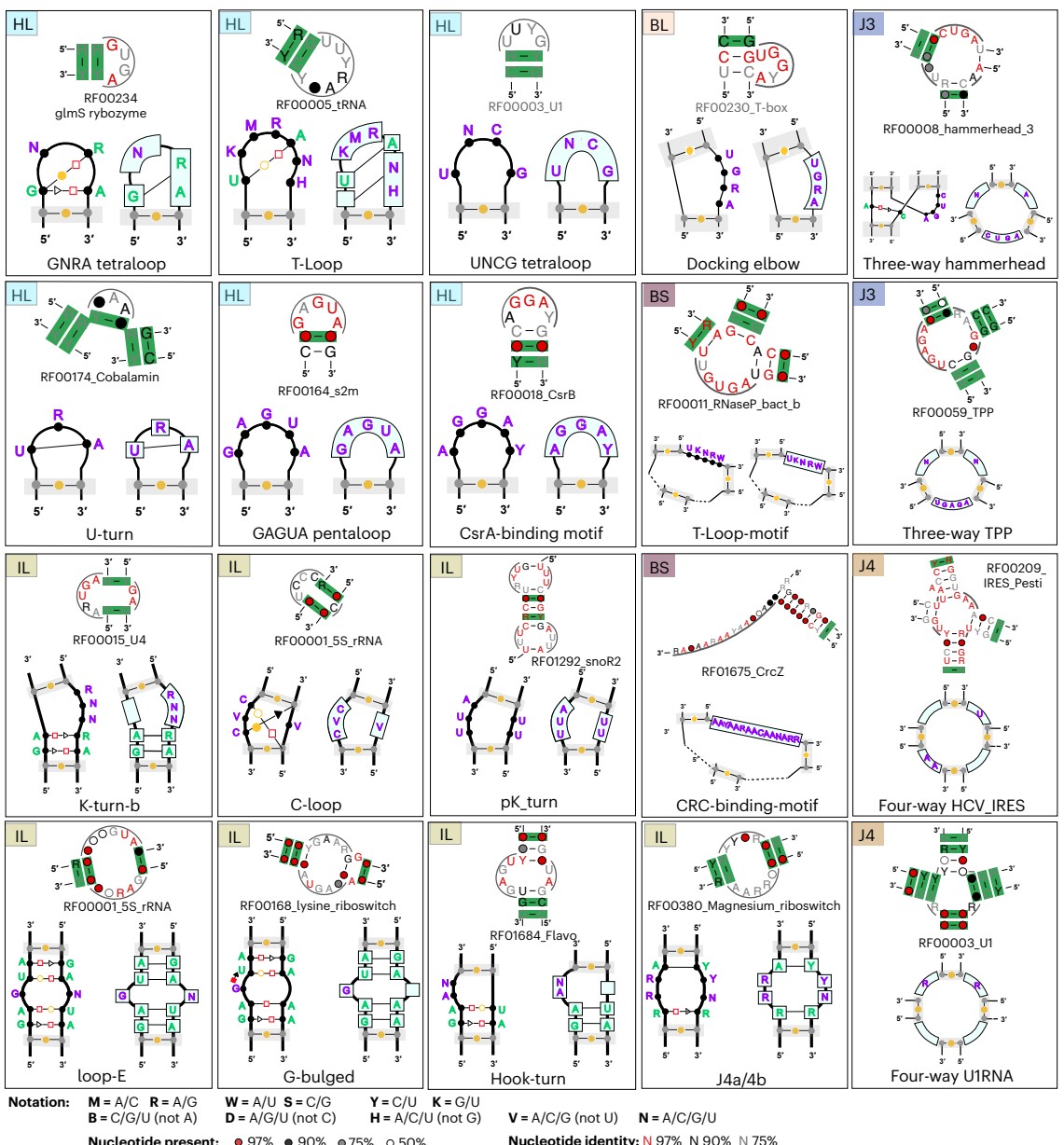

**Fig. 4 | Twenty RNA motifs with their R3D-grammar representation and detail from one Rfam family for which CaCoFold has identified a true instance of the motif with covariation support.** Hairpin loop motifs (HL): GNRA[79], T-loop[80], UNCG tetraloop[4,81,82]; U-turns[14], GAGAU pentaloop from conserved SARS region[83] and the CsrA binding motif[64]. Bulge motifs (BL): docking elbow[84]. Internal loop motifs (IL): K-turn-b[17] (a small variant of the K-turn in Fig. 3c), C-loop[85,86], Loop E[9,87],

G-bulge[88], pK-turn[89,90], Hook-turn[91] and J4a/4b internal loop[62]. J3 junction: the hammerhead ribozyme J3 junction[92] and the J3 junction of the TPP riboswitch[63]. J4 junction: in the hepatitis C virus internal ribosome entry site or HCV IRES[93] and in the U1 spliceosomal RNA[94]. Multiloop motif (BS): the T-loop domain[84] and the CRC binding domain[95]. Descriptors for the remaining 31 RNA 3D motifs included in the current version of CaCoFold-R3D are given in the Extended Data Fig. 3.

to be considered that could compromise the quality of the covariation signal. We have documented that pseudogenes and nonhomologous sequences weaken the covariation signal, and the use of structural alignments built on unconfirmed predicted structures introduces a circular reasoning that can produce spurious covariation[70]. We have shown that structural alignments of bona fide structural RNA sequences result in reliable 3D motif predictions informed by covariation found in alignments. Furthermore, even within a good structural alignment there is an important difference in confidence whether the motif prediction is bound by covariation or not.

## Performance comparison to other methods

In Extended Data Table 2, we present CaCoFold-R3D performance for an array of 3D motifs present in specific crystal structures.

With regard to comparison with the performance of related methods, RMDetect assesses 3D motif prediction in alignments for four internal loop motifs: the K-turn, C-loop, G-bulge and tandem GA motifs. The motifs are tested on SSU rRNA archaea and bacterial alignments (both of which include cases of all four motifs), SAM riboswitch and U4 RNA alignments (which include the K-turn motif) and a lysine riboswitch alignment (which includes the G-bulge motif). All those motifs are detected by CaCoFold-R3D in the corresponding Rfam alignments. The method BayesPairing2 assesses prediction sensitivity for two motifs (the sarcin−ricin and K-turn loops) tested on three families: LSU archaeal (RF02540) and LSU bacterial (RF02541) and the SAM riboswitch (RF00162). BayesPairing2 identifies the sarcin−ricin motif in the LSU archaeal and LSU bacterial and a K-tun in all three families. We identify the sarcin−ricin

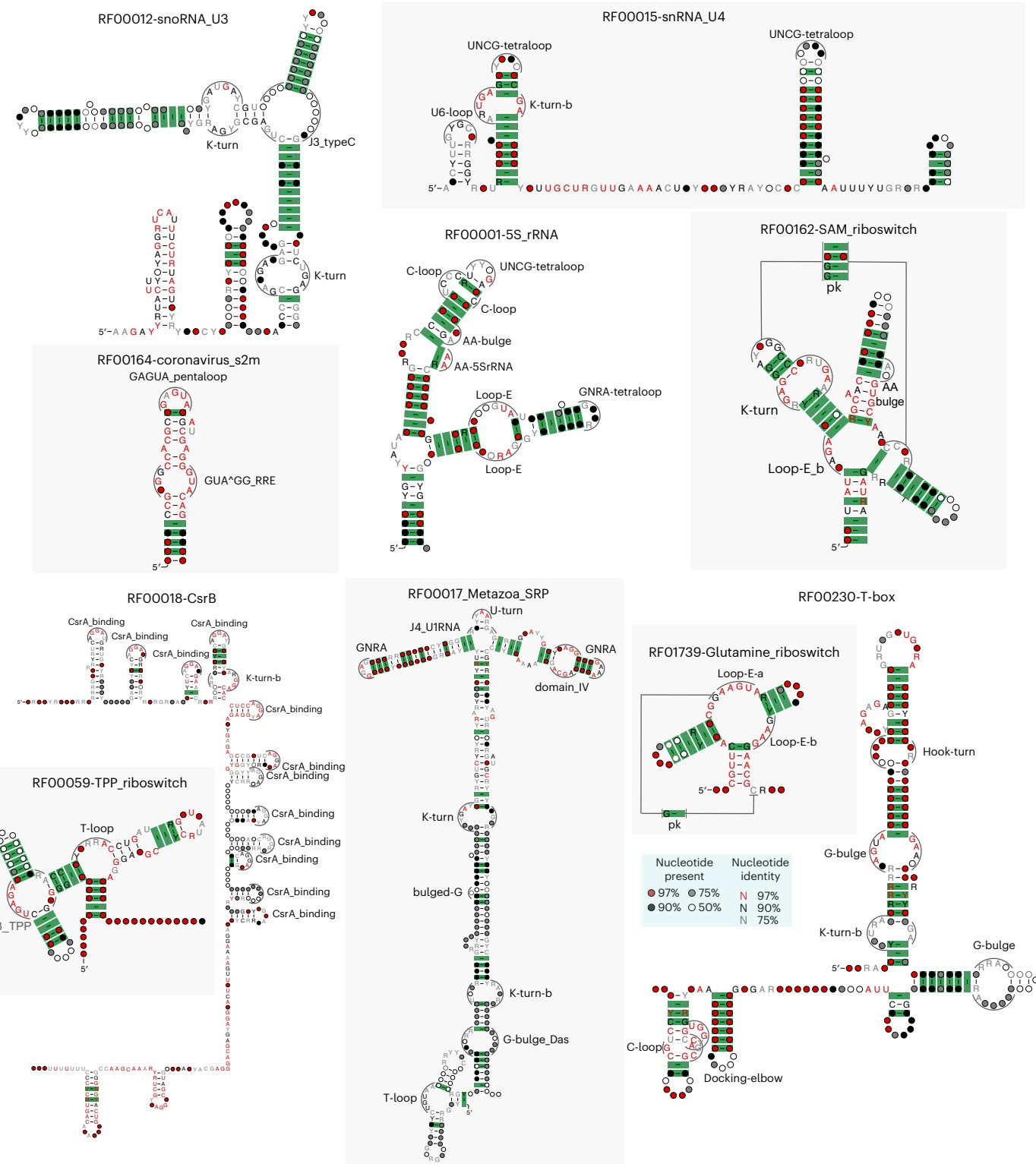

**Fig. 5 | CaCoFold-R3D structures confirmed by known 3D structures.** The examples of predicted consensus structures with 3D motifs are: the snoRNA U3[56], snRNA U4[58], the coronavirus stem-loop II motif (s2m)[83], the 5S rRNA[60], the SAM riboswitch[59], the CsrB RNA[64], the Metazoan SRP RNA with the domain IV motif[66], K-turn, U-turn[67] and T-loop[68], the Glutamine riboswitch[65], the T-box riboswitch[61],

and the TPP riboswitch[63]. CaCoFold-R3D uses a customized version of R2R[96] that automatically draws the RNA 3D motifs in the context of the rest of the consensus structure and its covariation. The collection of predictions for all the Rfam RNA families is provided in the supplemental materials.

motif in all four LSU families in Rfam (including LSU eukarya and LSU microsporidia), as well as in three other families. As for the K-turn motif, CaCoFold-R3D identifies it in the SAM riboswitch as well as in all four LSU families in Rfam. Notice that these results are obtained with one family-independent parameterization of CaCoFold-R3D

that generalizes to all families. This is unlike BayesPairing2, for which results vary a lot depending on which family has been used to train the model. For a more thorough comparison of CaCoFold-R3D's ability to detect known motifs relative to prior methods, see Extended Data Table 1.

CaCoFold-R3D not only detects all motifs mentioned in the literature of comparable methods but also detects many new motifs, especially internal loop and multiloop junction motifs, no other method has reported detecting.

### A new J3 junction motif with high representation

As an example of the power of CaCoFold-R3D as a tool to discover new motifs, we turn to a loop in the group II intron RNA for which Rfam describes a generic left bulge (Extended Data Fig. 4a). From the CaCoFold analysis of the Rfam seed alignment, we inferred that this is actually a J3 junction that is very conserved in sequence and exquisitely framed by covariation in all three closing helices (Extended Data Fig. 4b). Two of the closing helices are adjacent, and the third one is just a lone base pair. A group II crystal structure[71] confirms the coaxial stacking of the two adjacent helices, as well as the lone pair; it also reports one non-Watson–Crick base pair within the J3 junction (Extended Data Fig. 4c).

Because CaCoFold-R3D works on alignments, in the presence of a new potential motif, as in this case, the alignment already provides the potential consensus for the motif. Thus, the construction of a R3D descriptor is automatic. In this case, the consensus sequence profiles of the three branches of the J3 junction are (5′–3′) 'A', 'RAA' and empty.

We created the R3D grammar for this novel J3 motif (Extended Data Fig. 4d) and introduced it into the model. We were surprised to find that this seems to be a recurrent motif also found in other structural RNAs. In fact, our analysis shows that it is the most frequent J3 junction observed in Rfam as well as one of the top five most frequent motifs (Extended Data Table 1). In Extended Data Fig. 4e, we show examples of other J3-group II instances found in the CaCoFold-R3D structures for other Rfam families.

### Time performance

CaCoFold-R3D is fast. On an Apple M3 Max (128 GB), 98% of the Rfam families (4,079/4,178) take less than 60 s to run CaCoFold-R3D end-to-end, and 95% of families take less than 30 s. For the SSU and LSU of the rRNA—the two longest structured RNAs—it takes 32 min to analyze the eukaryotic SSU alignment (length 1,978 and 90 sequences) and 2.9 h for the eukaryotic LSU rRNA alignment (length 3,680 and 88 sequences).

Moreover, while other methods have to run a different search for each motif and for each sequence and also calculate a secondary structure separately[3,38], CaCoFold-R3D directly runs all 96 motifs together with the secondary structure in a single shot prediction and reports a consensus structure including 3D motifs for the alignment. The all-at-once RBGJ3J4-R3D prediction CYK algorithm scales with $\mathcal{O}(L^3 \times M)$ for an alignment (or sequence) of length $L$, where $M$ is the total number of nonterminals including both those for the RBGJ3J4 grammar (12) and those for the R3D grammars (96 in the tested implementation). Although, due to the covariation constraints, we expect this to be a worse-case behavior.

## Discussion

CaCoFold-R3D combines together several unique features that make the prediction of RNA 3D motifs accurate, fully integrated with secondary structure and annotated with their expected reliability. The R3D grammar abstracts the different 3D motifs into six generalized designs, unlocking the ability to incorporate an arbitrary number and variety of motifs—we provide results using a total of 96 motifs ('everything'). The RBGJ3J4 grammar specifies all possible loops in an RNA molecule, allowing motif detection in any possible location within a sequence ('everywhere'). CaCoFold-R3D is fully probabilistic, so one can compute the joint probability of all structural motifs together with all nested helices, pseudoknots and triplets ('all at once').

Because CaCoFold-R3D is flexible and easily customizable, we introduce a core of most representative 3D motifs including a new J3

junction. Because it is computationally fast, we are able to present full predictions for all Rfam families, including the ribosomal RNA. Finally, Because CaCoFold-R3D is framed by the evolutionary information contained in the alignment, it provides information on predictive confidence as a function of the number of significant covarying base pairs extracted from the input alignment. These results demonstrate that the R3D grammar coupled with covariation information offers an accurate and reliable prediction paradigm for identifying crucial 3D motifs in structural RNA sequences and an optimal tool to discover novel 3D motifs. CaCoFold-R3D will facilitate the construction of a database of structural elements present in RNA structures. This could achieve relevance similar to that of databases of protein domains[72].

Moreover, CaCoFold-R3D is a tool to provide valuable data to deep learning methods of RNA 3D structure prediction. Secondary structure is a crucial source of information to inform the prediction of RNA 3D structure in methods such as MC-Sym[46]. However, most methods ignore the rich information present in loops. Our method, able to predict jointly secondary and 3D motifs in one single predictive setting, opens the door for easily extracting this information and using it with methods trained on RNA sequences and alignments for a more robust RNA 3D structure prediction. In particular, CaCoFold-R3D predictions for the Rfam RNA families will be used to provide more complete inputs for the training of deep learning methods for RNA 3D structure prediction[73]. Methods that predict RNA 3D structure such as AlphaFold3[74] and RoseTTaFold All-Atom (RFAA)[75], which already use the Rfam data to inform their inputs, will benefit from the comprehensive information contained on the prevalent 3D recurrent motifs present in all RNA 3D structures provided by CaCoFold-R3D.

CaCoFold-R3D has the potential to have an impact in small molecule drug therapeutics that target RNA binding sites. RNA therapeutic compounds such as risdiplam (for spinal muscular atrophy)[76] rely of the presence of RNA loops forming favorably binding pockets[77,78]. Our method is proven to hold important predictive information about key structural elements within RNA loops, which could greatly help guide the design of RNAs whose loop configurations serve as therapeutic targets for small molecule drugs.

## Online content

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

## Methods

### The RBG grammar including J3 and J4 junctions (RBGJ3J4)

The RBG[44] is a probabilistic SCFG modeling RNA folding, targeting secondary structural elements including stacked canonical base pairs forming helices, hairpin loops, bulge and internal loops, and multiloops.

SCFGs build correlated states at arbitrary distances and hence are well suited for expressing RNA base pairing. Moreover, the nonterminals directly correspond to the overarching secondary structures. For example, the nonterminal 'F' builds a helix, while the nonterminal 'P' is present after a helix has ended and is able to initiate all possible loop features such as a hairpin loop, internal loop or multiloops (Extended Data Fig. 1).

Because many RNA 3D motifs occur in multiloops, especially the most frequent J3 and J4 junctions, here we introduce the RBGJ3J4 grammar, which singles out J3 junctions (multiloops framed by three helices) as well as J4 junctions (multiloops framed by four helices) as specific cases from any other higher-order multiloop (see Fig. 2a and Extended Data Fig. 1 for the full description of the RBGJ3J4 grammar).

Another unique feature of the RBG and RBGJ3J4 grammars as they are used in CaCoFold is that they fold an alignment, not a single sequence. A position does not represent one residue but a probability vector describing the frequency of each residue in the aligned column. This way, we can produce consensus secondary structures that include the information contained in all aligned sequences.

### The six types of R3D grammars

**Hairpin loop motifs.** The 3D motifs in hairpin loop motifs include both residues paired through non-Watson–Crick interactions as well as unpaired ones. For instance, the GNRA tetraloop[79] is a frequent hairpin loop motif in which the first G base forms two non-Watson–Crick interactions with the R and A bases, which provides extra thermodynamic stability to the tetraloop[79]. The GNRA R3D SCFG models the correlated occurrence of the G and the NA base pairs (but it does not model the type of base pairing involved in that correlation), as well as the unpaired N residue (Fig. 3a).

R3D designs a generic hairpin loop 3D motif by an arbitrary number of left/right correlated segments and a final loop segment of residues not correlated elsewhere. Figure 3a shows the general model. The R3D-hairpin loop motif assigns a profile HMM to the loop sequence, as well as to all the allocated left and right segments which will consist of the contiguous subsequences that pair through non-Watson–Crick interactions.

**Bulge loop motifs.** R3D bulge loop motifs are described in Fig. 3b, and they have similar properties to the hairpin loop motifs. Notice that a bulge loop motif can appear in a left of right bulge depending on which of the two ends of the motif is continuous and which inserts itself with the rest of the structure. Figure 3b shows only one of the two possibilities (called variants) for the bulge loop motif. We generalize the concept of motif variants in the following sections and Extended Data Fig. 2.

**Internal loop motifs.** For an internal loop motif (Fig. 3c), R3D assumes the presence of two loop regions with an inner stem and an outer stem region which are emitted correlatedly by the SCFG. As with the hairpin loop motifs, the actual sequences in the loops and left/right inner and outer stem sequences (all of which can be potentially empty) are modeled by profile HMMs.

For instance, the K-turn (or Kink turn) is a common internal loop motif featuring two G–A hydrogen bonded Sugar–Hoogsteen edge interactions that help induce an axial bend[17]. The K-turn R3D SCFG models these two correlated interactions. The internal loop portion of the K-turn has three unpaired nucleotides with consensus RNN, so the R3D grammar adds a profile HMM for the right bulge RNN sequence and treats the left bulge as empty (Fig. 3c).

**J3 and J4 junction motifs.** The R3D SCFG for a J3 motif includes three sequence segments that are emitted correlatedly (Fig. 3d). For instance, for the Hammerhead ribozyme, while R3D does not model the non-canonical base pairs that occur within the junction, it does model the correlated emission of all three segments which include the base paired residues as well as those that are not paired but part of the motif. Similarly, for a J4 junction motif four arbitrary sequence segments are considered simultaneously (Fig. 3e). As seen in the case of the HCV IRES J4 junction, the correlated segments may include no nucleotides, thus indicating helix coaxial stacking[97].

**Other multiloop motifs.** The R3D grammar also introduces sequence (branch segment) motifs which can appear in any multiloop branch as described in Fig. 3f. These motifs may describe particular protein-binding motifs such as the CsrA binding motifs of the CsrB RNA[64], as well as components of higher-order loop motifs. For instance, the loop E that appears in a J3 junction of the glutamine riboswitch[65], which is interrupted by a one base pair pseudoknot, and R3D is able to model with two branch segment motifs.

### The sequence-motif profile HMMs

Each interacting partner or loop in a RNA 3D motif consists normally of a conserved sequence with some variability. R3D models those sequence segments as short profile HMMs described in Fig. 3g. Each profile HMM has a consensus sequence, and by allowing mutations, insertions and deletions, it is able to accommodate sequence variability and to identify motif instances that have some variability relative to the consensus. The states of the profile HMM emit on transition, not on state. Motifs with sequence segments without residues, such as those occurring in multiloops bounded by coaxially stacked helices, are also possible. We model empty segments with a profile HMM to permit the possibility of insertions relative to consensus.

Each sequence motif is characterized by a consensus sequence $S_1 \ldots S_L$, and it gets assigned a profile HMM (Fig. 3g). The profile HMM introduces one consensus state per position $S_i$, for $1 \le i \le L$. Each consensus state is characterized by a consensus residue or residue type such as: A, or R (A or G), or Y (C or U) or others that determine the emission probability of the consensus. There is an error probability to allow any of the other residues not designated by the consensus. That is a $S = R$ position assigns $P_S(A) = P_S(G) = 0.5 - \epsilon/2$ and $P_S(C) = P_S(U) = \epsilon/2$. The profile HMM also includes one insertion states per position. All inserted position use the same emission residue probability distribution matching the residue frequencies of the training set. Emissions are done on transition, and transitions without emissions describe deletion events.

Each profile HMM is parameterized by a length distribution with an expected length closely matching that of the consensus motif, and allowing insertions and deletions relative to consensus. Empty sequence motifs are also modeled by a profile HMM to permit the possibility of more divergent motif examples with insertions.

The profile HMMs described in Fig. 3g can be used to score either individual sequence positions or alignment positions. Introducing a probability distribution per position $o$, $\{p_o(a)\}_{a=A,C,G,U/T}$, the consensus and indel emission probabilities of the profile HMM can be written as

$$P_{S_i}(o) = \sum_{a=A,C,G,U} p_o(a) P_{S_i}(a),$$

$$P_{indel}(o) = \sum_{a=A,C,G,U} p_o(a) P_{indel}(a).$$

For the case of an aligned position, $p_o$ is the position base composition. For the case of a particular sequence, there is a single residue per position $o = a$, such that $p_o(o = a) = 1$, and

$$P_{S_i}(o = a) = P_{S_i}(a),$$

$$P_{indel}(o = a) = P_{indel}(a)$$

For each tested segment (in alignment or sequence), the profile HMM calculates the probability of the segment given the motif using the forward algorithm. The probability for a given subsequence is incorporated into the corresponding R3D SCFG where the sequence segment is included.

### The RBGJ3J4-R3D joint grammar uses the CYK folding algorithm

The R3D motif SCFGs are integrated into the RBGJ3J4-R3D grammar as described in Fig. 2. CaCoFold-R3D admits an arbitrary number of 3D motifs. Extended Data Fig. 3 describes the list of 51 motifs used in this manuscript. CaCoFold-R3D internally interprets the descriptor lists and implements a SCFG for each motif according to the general R3D grammars for each of the six types of motif as described in Fig. 3.

RNA 3D motifs bound by two or more helices can appear in different configurations depending on which of the ends corresponds to the 5′/3′ ends of the molecule, versus all the others for which the backbone is continuous. That is, bulge loops and internal loops can have two configurations, while J3s and J4s can have three and four, respectively (Extended Data Fig. 2). For a given representation of the motif, R3D internally implements and adds all possible configurations which get added to the RGBJ3J4-R3D grammar.

As with the RBGJ3J4 grammar, all R3D SCFGs describe a consensus motif in an alignment, not just a particular sequence motif. Thus, R3D is able to represent the variability observed in the motif. The input to the algorithm is not one nucleotide per sequence position, but $L$ probability distributions of dimension 4 describing the frequency of each nucleotide per alignment position (a probabilistic sequence) and $L \times L$ distributions describing join $4 \times 4$ pair probabilities.

RBGJ3J4-R3D implements the CYK folding algorithm[98,99] to report the consensus fold and consensus 3D motifs that maximize the probability of the alignment. This CYK algorithm is applied not to one single sequence but to a probability sequence where for each position, there is not one given residue but a probability distribution representing the frequency of each nucleotide in each alignment position. This generalized CYK on probability sequences has the same algorithmic complexity as the original CYK on individual sequences.

### The CaCoFold-R3D multilayered prediction method

CaCoFold-R3D uses a multilayered folding approach constrained by covariation. The first layer includes a maximal set of nested covarying base pairs and uses the RBGJ3J4-R3D CYK algorithm introduced here, which replaces the RBG CYK algorithm of CaCoFold[45]. This first layer folding identifies the main secondary structure together with any 3D motif to produce a structure with the maximum probability using the computationally efficient CYK algorithm.

The treatment of the higher layers of CaCoFold-R3D is similar to CaCoFold. When the covariation found in the alignment by R-scape cannot be fully explained by a nested structure, those non-nested covarying pairs are incorporated in additional folding layers. The number of layer depends on the number additional clusters of nested covarying pairs that can be constructed. These subsequent folding layers of CaCoFold-R3D use a simpler folding grammar (named G6XS and described in ref. 45) to introduce additional helices supported by covariation that could be representing any combination of pseudoknots, alternative helices or triplet interactions with covariation support. In that form, CaCoFold-R3D is able to identify pseudoknotted helices as folds appearing at subsequent layers along with any 3D motif that the first CYK layer may have already identified for that loop.

As an example, the glutamine riboswitch structure presented in Fig. 5 identifies a pseudoknot (one single base pair with covariation support) and a loop-E 3D motif in the same J3 junction; both motifs are confirmed by the crystal structure[65].

There may be situations in which the secondary structure or the 3D motif annotation reported by the first layer overlaps with a pseudoknot or triplet reported by the higher layers. Those could represent conflicts or they could be alternative structures. We leave any of those unmodified for the user to see and make further assessments.

### Availability
The source code can be downloaded via GihHub at https://github.com/EddyRivasLab/R-scape, as well as from the Supplementary Information associated to this manuscript.

### Reporting summary
Further information on research design is available in the Nature Portfolio Reporting Summary linked to this article.

## Data availability
We used Rfam v15.0 (ref. 52). The data used from Rfam is provided as part of the Supplementary Information, and it can also be downloaded from https://ftp.ebi.ac.uk/pub/databases/Rfam/CURRENT/Rfam.seed.gz.

## Code availability
We introduce CaCoFold-R3D in the software package R-scape v2.5.7, which is provided as part of the Supplementary Information, and can also be downloaded via GitHub at https://github.com/EddyRivasLab/R-scape/blob/master/versions/rscape/rscape_v2.5.7.tar.gz. The R3D-prototype python code is provided as part of the Supplementary Information. We used TORNADO v0.6.0 (ref. 51), which can be downloaded via GitHub at https://github.com/EddyRivasLab/tornado/blob/master/versions/tornado_v0.6.0.tar.gz.

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

## Acknowledgements

We thank E. Westhof and V. Reinharz for insights on the RNA 3D motifs. We thank L. Merk for bringing to our attention the J3 junction motif in the group II intron RNA. We thank S. Eddy for a critical reading of the manuscript. This work was supported by NIH grant no. R01-GM144423

to E.R. A.K. was funded by a Harvard Undergraduate Research Fellowship and a Paul and Daisy Soros Fellowship.

## Author contributions

E.R. conceived the research. A.K. and E.R. designed the algorithms. A.K. implemented the python R3D-prototype. E.R. implemented the CaCoFold-R3D method. A.K. and E.R. wrote the manuscript.

## Competing interests

The authors declare no competing interests.

## Additional information

**Extended data** is available for this paper at https://doi.org/10.1038/s41592-025-02833-w.

**Correspondence and requests for materials** should be addressed to Elena Rivas.

# RNA Basic Grammar (RBG)

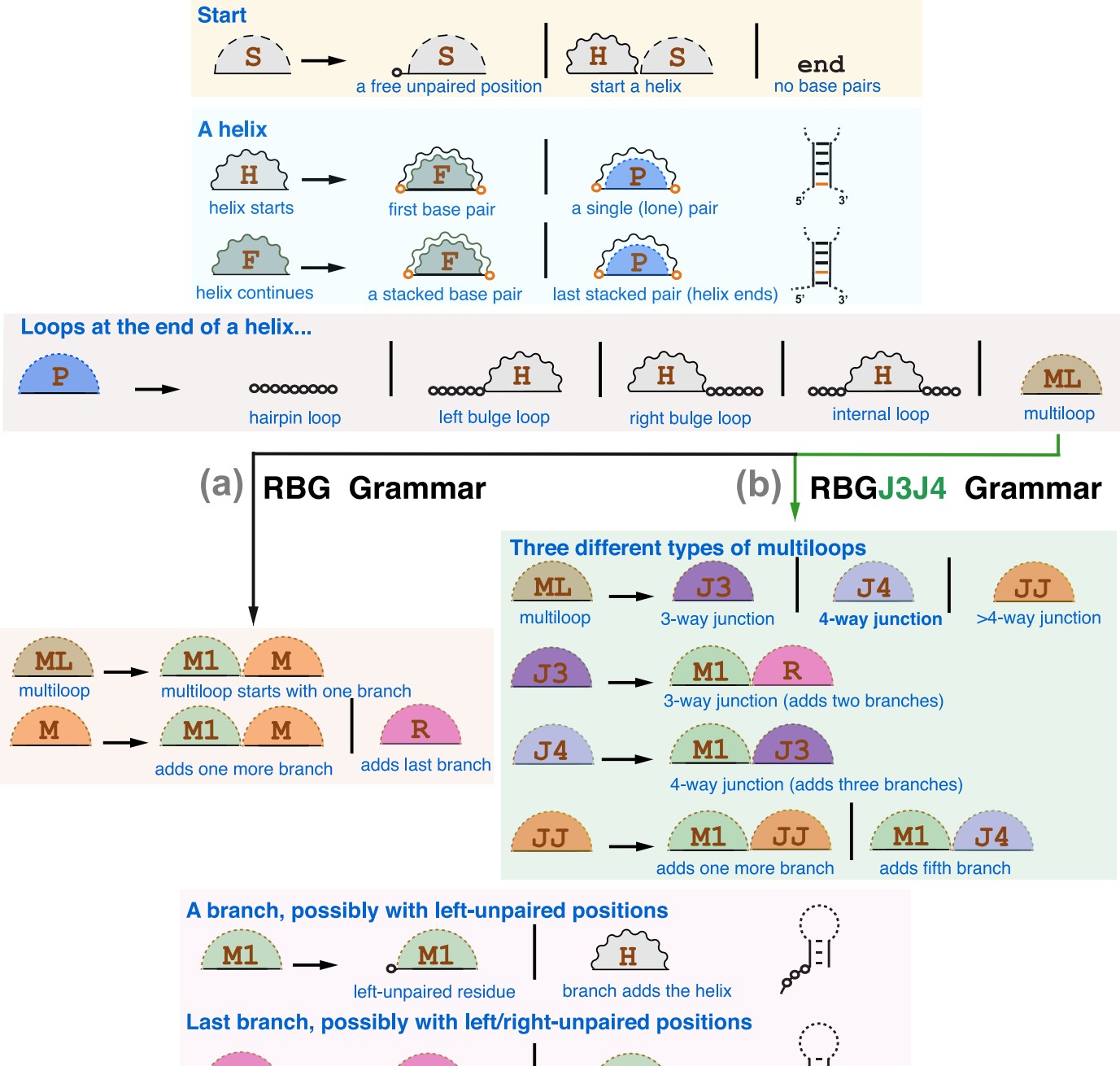

**Extended Data Fig. 1 | The RBG and RBGJ3J4 generative grammars.** Inserts (**a**) and (**b**) describe the two distinct realizations of the multiloop non-terminal ML. RBGJ3J4 replaces the generic multiloop non-terminal M with non-terminals J3, J4 and JJ in order to distinguish 3-way and 4-way junctions from other higher order multiloops. A solid line represent the RNA sequence, a curly line indicates that the two connecting residues are base paired, and a dashed line indicates that the relationship between the two residues is yet undetermined. Non-terminals are depicted in brown and actual residues/positions are depicted with circles (black for unpaired and orange for base paired positions). Each non-terminal describes a discrete random variable of events which are enumerated on the right-hand side of the arrows. The allowed events for a given non-terminal are separated by the | ('or') symbol. Starting from the S non-terminal, an RNA sequence/structure is produced by sampling from the discrete probability distributions (transitions and emissions) associated to each non-terminal.

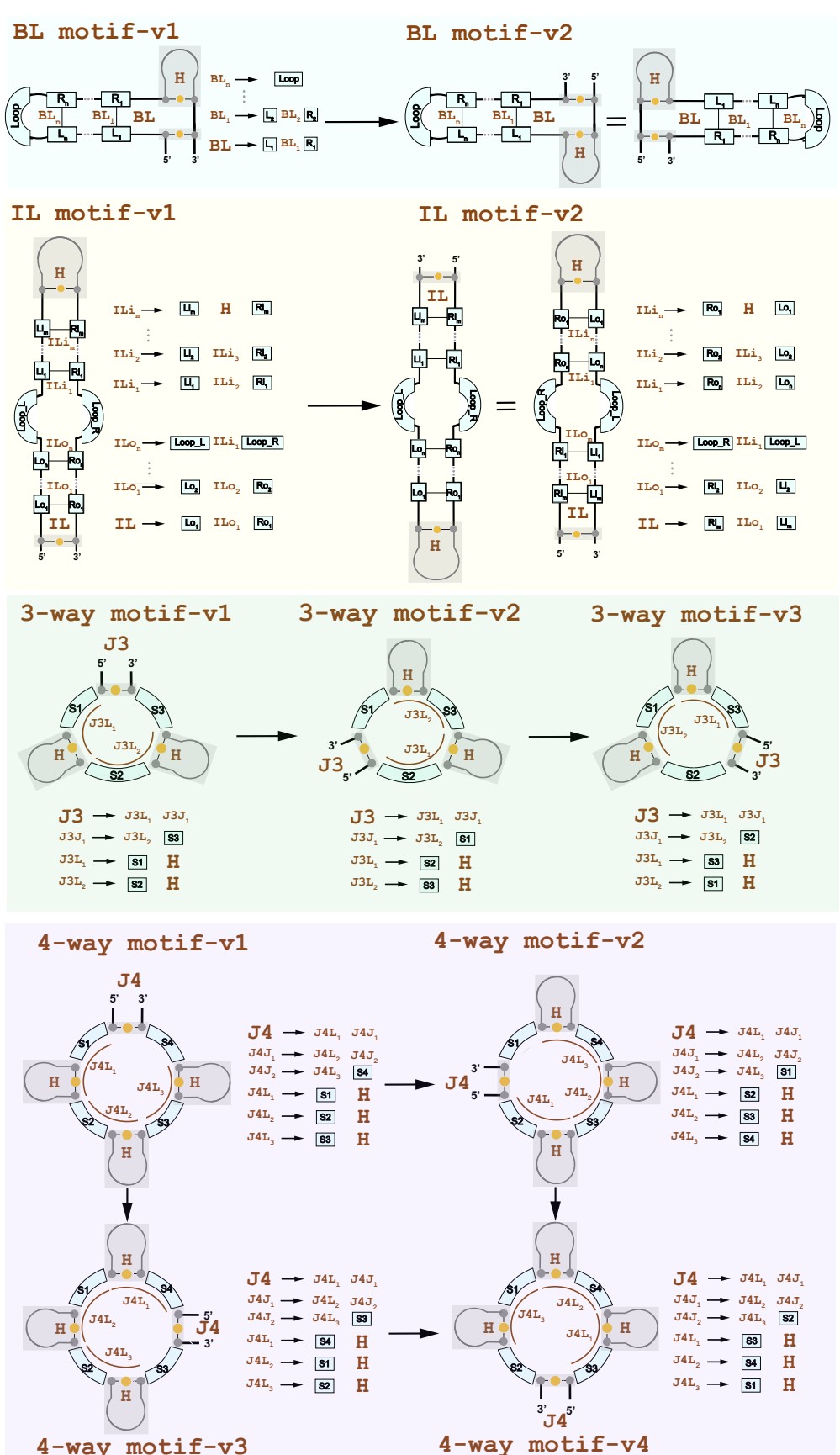

**Extended Data Fig. 2 | RNA 3D motifs variants.** RNA 3D motifs bound by more than one helix (*that is* all except for hairpin loops) allow different topological variants depending on which 5′/3′ ends are selected to integrate the motif into the rest of the structure. Bulge and Internal loop motifs have two variants, and 3-way and 4-way junctions have three and four variants respectively. For any 3D motif entry, CaCoFold-R3D internally models all possible variants of the motif.

```
#HL   Loop(5'-3')  L(5'-3')    R(5'-3')      name
#----------------------------------------------------------------------------
 HL   N            G           RA            GNRA-tetraloop
 HL   URA          -           -             U-turn
 HL   UNCG         -           -             UNCG-tetraloop
 HL   ANYA         -           -             ANYA-tetraloop
 HL   CUYG         -           -             CUYG-tetraloop
 HL   YGNN         -           -             YGNN-tetraloop
 HL   GANC         -           -             GANC-tetraloop
 HL   UNAC         -           -             UNAC-tetraloop
 HL   URRR         -           -             T-loop-tetraloop
 HL   UAACR        -           -             L8_RNaseP_bact_a
 HL   AG           UAGUACG     AGGACC        Sarcin-ricin_loop
 HL   AGGAY        -           -             CsrA_binding
 HL   GAGUA        -           -             GAGUA_pentaloop
 HL   AA           -           -             AA-5SrRNA
 HL   GCRYA        -           -             U6-loop

#BL   Loop(5'-3')  L(5'-3')    R(5'-3')      name
#----------------------------------------------------------------------------
 BL   UGRAA        -           -             Docking-elbow
 BL   AA           -           -             AA-bulge
 BL   G            -           -             bulged-G

#IL   Loop-L(5'-3') Loop-R(5'-3') L-o(5'-3')  R-o(5'-3')   L-i(5'-3')  R-i(5'-3')   name
#----------------------------------------------------------------------------
 IL   -             RNN          N,G,A        N,A,R         -           -            K-turn
 IL   -             RNN          G,A          A,R           -           -            K-turn-b
 IL   G             N            G,A          A,U           U,A         A,G          Loop-E
 IL   UAAG          UAU          C,C          G,G           -           -            GAAA_Tetraloop-receptor
 IL   CVC           V            -            -             -           -            C-loop
 IL   G             -            G,A          A,A           U,A         A,G          G-bulge
 IL   G             -            U,A          C,C           U,A         A,G          G-bulge_Das
 IL   -             -            G,A          A,G           -           -            Tandem-GA
 IL   YCC           AAC          -            -             -           -            Twist-up
 IL   YAA           RAN          -            -             -           -            UAA_GAN
 IL   RR            YN           R            R             A           Y            J4a/4b
 IL   AA            AA           -            -             -           -            J4/5-IL
 IL   GUA           GG           -            -             -           -            GUA^GG_RRE
 IL   CAGG          AGCA         -            -             -           -            S_domain
 IL   AN            -            G,A          A,U           -           -            Hook-turn
 IL   UGRAA         -            -            -             -           -            Docking-elbow-IL
 IL   UU            AUU          -            -             -           -            pK-turn

#J3   S1(5'-3')    S2(5'-3')   S3(5'-3')     name
#----------------------------------------------------------------------------
 J3   N            CUGA        A             J3_hammerhead
 J3   U            YUCUAC      AC            J3_purine
 J3   NN           NNNNNN      NN            J3_typeA
 J3   NNNN         NNNN        NNNN          J3_typeB
 J3   NN           NNN         NNNNNN        J3_typeC
 J3   N            UGAGA       N             J3_TPP
 J3   A            RAA         -             J3_groupII

#J4   S1(5'-3')    S2(5'-3')   S3(5'-3')     S4(5'-3')     name
#----------------------------------------------------------------------------
 J4   -            AA          -             U             J4_HCV_IRES
 J4   N            -           NNN           -             J4_tRNA
 J4   N            N           NN            -             J4_manA
 J4   R            -           -             R             J4_U1RNA

#BS   Loop(5'-3')       name
#----------------------------------------------------------------------------
 BS   UKNRW             T-loop
 BS   RRGU              LoopE-a
 BS   RARR              LoopE-b
 BS   AAYAARAACAANARR   CRC_binding
 BS   AGGAY             CsrA_motif
```

**Extended Data Fig. 3 | RNA 3D motifs descriptors.** The descriptor file includes 51 different distinct motifs. CaCoFold internally constructs SCFGS for a total of 96 motif variants. The 96 motif R3D SCFGs get integrated into the RBGJ3J4 grammar. The RBGJ3J4-R3D grammar folds and detect the motifs of the RNA simultaneously. HL = Hairpin Loop, BL = Bulge Loop, IL=Internal Loop, J3 = 3-way Junction, J4 = 4-way Junction, BS = Branch Segment.

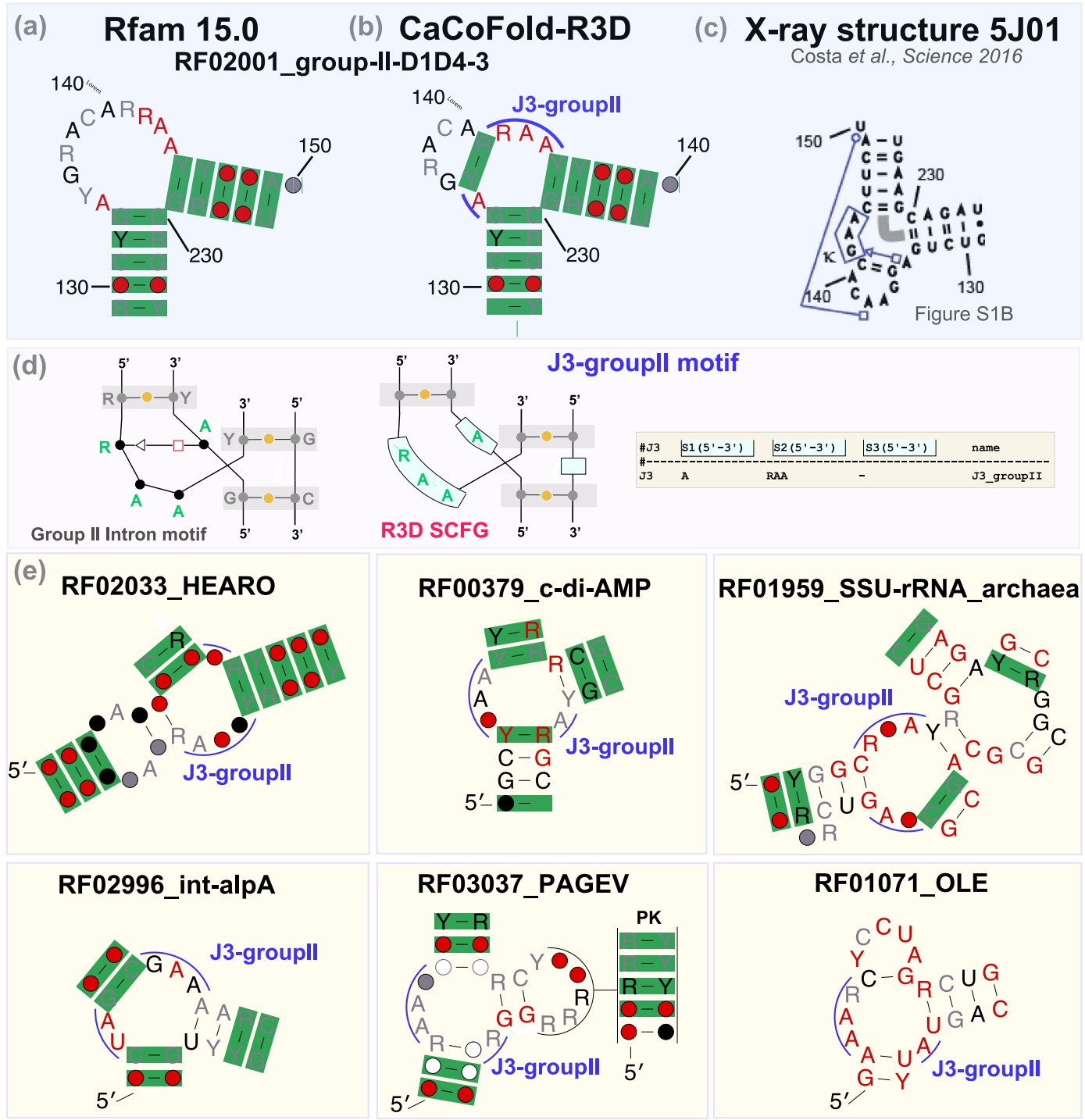

**Extended Data Fig. 4 | Group II intron 3-way junction motif. (a)** Detail of a bulge described by the Rfam structure for the Group II intron RNA. (Coordinates are for reference to the crystal structure in **c**). (**b**) 3-way junction identified by CaCoFold with covariation support in all three helices. Two of the helices are adjacent, and one is a lone base pair. (**c**) Detail of a Group II intron crystal structure which confirms the presence of the 3-way junction, a coaxial stacking between two of the helices and the lone Watson-Crick base pair, as well as a non Watson-Crick base pair in the 3-way junction. (**d**) Description of the J3-groupII motif and its R3D model. (**e**) Examples of CaCoFold-R3D predictions for other Rfam families that also include the J3-groupII junction.

## Extended Data Table 1 | Non-canonical motif de novo prediction from alignments

| Type | Motif name | PDB | Rfam homology | CaCoFold-R3D | Other Methods | Motif Model |
|---|---|---|---|---|---|---|
| HL | GNRA-tetraloop | 7EI6 | synthetic | 170 | – | Hendrix *et al.*, 2005 [18] |
| HL | U-turn | 4WFL | RF00017_Metazoan_SRP | YES | – | Kempf *et al.*, 2014 [23] |
| HL | UNCG-tetraloop | 1F7Y | synthetic | 48 | – | Hendrix *et al.*, 2005 [18] |
| HL | ANYA-tetraloop | 6MSF | synthetic | 18 | – | Hendrix *et al.*, 2005 [18] |
| HL | CUYG-tetraloop | 2L6I | synthetic | 20 | – | Hendrix *et al.*, 2005 [18] |
| HL | YGNN-tetraloop | 1AFX | synthetic | 13 | – | Hendrix *et al.*, 2005 [18] |
| HL | GANC-tetraloop | 3BWP | synthetic | 15 | – | Keating *et al.*, 2009 [22] |
| HL | UNAC-tetraloop | 4A4R | synthetic | 26 | – | Zhao *et al.*, 2012 [100] |
| HL | T-loop-tetraloop | 7SAM | RF01085_TLS-PK4 | YES | – | RMfam [37] |
| HL | L8_RNaseP_bact_a | 1U9S | RF00010_RNaseP_bact_a | YES | – | Das & Baker, 2010 [21] |
| HL | Sarcin-Ricin_loop | 1SCL | RF02540_LSU_rRNA_archaea | YES | BayesPairing2 | Szewczak *et al.*, 1993 [101] |
| HL | CsrA_binding | 2MF0 | RF00018_CsrB | YES | – | Duss *et al.*, 2014 [24] |
| HL | GAGUA_pentaloop_sars | 1XJR | RF00164_Coronavirus s2m motif | YES | – | Das & Baker, 2010 [21] |
| IL | Kink-turn_SAM_riboswitch | 2GIS | RF00162_SAM_riboswitch | YES | BayesPairing2 | Das & Baker, 2010 [21] |
| IL | Kink-turn_rRNA | 1JJ2 | RF02540_LSU_rRNA_archaeal | YES | RMDetect | Das & Baker, 2010 [21] |
| IL | LoopE | 354D | RF00001_5S_rRNA | YES | – | Das & Baker, 2010 [21] |
| IL | GAAA_Tetraloop-receptor | 1GID | RF00028_Group I intron | YES | – | Cate *et al.*, 1996 [8] |
| IL | C-loop | 1S72 | RF02541_LSU_rRNA_bacterial | YES | RMDetect | Lescoute *et al.*, 2005 [85] |
| IL | G-Bulge | 3DIL | RF00168_Lysine riboswitch | YES | RMDetect | Serganov *et al.*, 2008 [88] |
| IL | G-Bulge_Das | 1Q9A | RF02541_Bacterial LSU_RNA | YES | – | Das & Baker, 2010 [21] |
| IL | tandem GA | 1SA9 | RF00177_SSU_rRNA_bacteria | YES | RMDetect | Cruz & Westhof, 2011 [3] |
| IL | Twist up | 1J5E | RF02541_LSU_rRNA_bacterial | NO, 9 | – | Zhong & Zhang, 2011 [102] |
| IL | UAA/GAN | 6AHR | RF00009_RNaseP_nuc | YES | – | Lee *et al.*, 2006 [103] |
| IL | J4a/4b | 2QBZ | RF00380_M-box_riboswitch | YES | – | Das & Baker, 2010 [21] |
| IL | J4/5 | 2R8S | RF00028_Group I intron | NO, 6 | – | Das & Baker, 2010 [21] |
| IL | GUA/GG_RRE | 1CSL | RF00055_HIV-1 RRE | too conserved, 10 | – | Das & Baker, 2010 [21] |
| IL | SRP RNA DOMAIN IV | 1CQ5 | RF00017_Metazoan_SRP | YES | – | Schmitz *et al.*, 1999 [66] |
| IL | Hook_turn_GAAC/CUA | 1MHK | synthetic | 32 | – | Das & Baker, 2010 [21] |
| IL | Docking elbow-IL | 3Q1Q | RF00011_RNaseP_bact_b | YES | – | Lehmann *et al.*, 2012 [84] |
| IL | pK-turn | 3OK7 | RF00010_RNase P RNA | NO, 45 | – | Meyer *et al.*, 2012 [90] |
| J3 | Purine_riboswitch_J3 | 2EEW | RF00167_Purine riboswitch | YES | – | Das & Baker, 2010 [21] |
| J3 | hammerhead_CUGA/A/C | 359D | RF02275_Hammerhead_HH9 | YES | – | Das & Baker, 2010 [21] |
| J3 | J3_typeA | 4RZD | RF02680_PreQ1-III | YES | – | Lescoute & Westhoff, 2006 [104] |
| J3 | J3_typeB | 1J5E | RF00177_SSU_rRNA_bacteria | YES | – | Lescoute & Westhoff, 2006 [104] |
| J3 | J3_typeC | 4LX6 | RF03165_2dG-II riboswitch | YES | – | Lescoute & Westhoff, 2006 [104] |
| J3 | J3_TPP | 2GDI | RF00059_TPP riboswitch | YES | – | Serganov *et al.*, 2006 [63] |
| J3 | J3_groupII | 5J01 | RF02001_group-II-D1D4-3 | YES | – | this work |
| J4 | HCV_IRES_AA/0/U/0 | 1KH6 | RF00061_IRES_HCV | YES | – | Das & Baker, 2010 [21] |
| J4 | J4_U1RNA | 3PGW | RF00003_U1 RNA | YES | – | Weber *et al.*, 2010 [16] |
| BS | T-loop | 1EHZ | RF00005_tRNA | YES | – | Hendrix *et al.*, 2005 [18] |
| BS | LoopE-a | 4FRN | RF01689_AdoCbl-II | YES | – | this work |
| BS | LoopE-b | 4FRN | RF01689_AdoCbl-II | YES | – | this work |
| BS | CRC_binding | – | RF00195_RsmY | YES | – | Sonnleitner *et al.*, 2009 [105] |
| PK | | 5DDP | RF01739_Glutamine riboswitch | YES | – | Ren *et al.*, 2015 [65] |

For each motif, we identify one crystal structure that includes the motif, and we test for the prediction of the motif on the corresponding Rfam alignment. We also report other methods that have identified the motif in the same family. For motifs associated to a crystallized synthetic oligo, or not identified by CaCoFold-R3D, we provide the number of other Rfam families that include that motif with covariation support. The specifications of the motifs are provided in the R3D descriptor file given in Extended Figure 3. CaCoFold-R3D results for the Rfam seed alignments are given in the supplemental materials file `results/CaCoFold-R3D/CaCoFold-R3D_Rfam`.

**Extended Data Table 2 | RNA 3D motifs found in Rfam**

RFAM SEED ALIGNMENTS
4,178 Families

| Type | Motif | # Motifinstances with cov support (all) | # Rfam families with motif with cov support (all) | SSU eukarya with support (all) | LSU eukarya with support (all) |
|---|---|---|---|---|---|
| HL | GNRA_tetraloop | 170 (214) | 101 (132) | 4 (4) | 6 (7) |
| IL | K_turn | 68 (83) | 54 (68) | 2 (2) | 2 (2) |
| BL | Docking_elbow | 59 (94) | 55 (83) | 2 (2) | 2 (5) |
| J3 | J3_groupII | 52 (57) | 40 (43) | 1 (1) | 1 (1) |
| HL | UNCG_tetraloop | 48 (78) | 42 (69) | 1 (1) | 0 (0) |
| J3 | J3_typeA | 46 (49) | 36 (39) | 4 (4) | 4 (4) |
| BL | AA_bulge | 45 (101) | 34 (81) | 1 (1) | 2 (2) |
| IL | pK_turn | 45 (82) | 41 (75) | 0 (0) | 0 (0) |
| J3 | J3_typeC | 43 (45) | 42 (43) | 0 (0) | 1 (1) |
| J4 | J4_U1RNA | 40 (44) | 29 (31) | 0 (0) | 3 (4) |
| IL | UAA_GAN | 40 (50) | 23 (33) | 0 (0) | 5 (5) |
| IL | G_bulge | 37 (52) | 30 (39) | 0 (0) | 1 (1) |
| IL | K_turn_b | 35 (50) | 32 (47) | 0 (1) | 1 (1) |
| J4 | J4_HCV_IRES | 33 (36) | 25 (28) | 3 (3) | 1 (1) |
| IL | Hook_turn | 32 (49) | 28 (45) | 0 (0) | 3 (3) |
| IL | Tandem_GA | 29 (37) | 19 (25) | 2 (3) | 3 (3) |
| BS | LoopE_a | 28 (33) | 15 (20) | 1 (1) | 4 (4) |
| BS | T_loop | 27 (33) | 25 (30) | 1 (1) | 2 (2) |
| J4 | J4_manA | 26 (27) | 24 (25) | 0 (0) | 2 (2) |
| IL | C_loop | 26 (42) | 25 (40) | 1 (1) | 0 (0) |
| BS | CsrA_motif.rev | 26 (30) | 23 (27) | 2 (2) | 1 (1) |
| HL | UNAC_tetraloop | 26 (48) | 26 (47) | 0 (0) | 1 (1) |
| BS | CRC_binding | 26 (34) | 18 (22) | 5 (5) | 0 (0) |
| J4 | J4_tRNA | 25 (26) | 19 (20) | 3 (3) | 3 (3) |
| HL | T_loop_tetraloop | 24 (43) | 20 (34) | 1 (3) | 0 (0) |
| BS | LoopE_b.rev | 24 (25) | 18 (18) | 1 (1) | 1 (1) |
| BS | LoopE_b | 21 (22) | 15 (16) | 1 (1) | 2 (2) |
| BL | bulged_G | 20 (43) | 18 (37) | 0 (0) | 0 (0) |
| BS | T_loop.rev | 20 (26) | 17 (22) | 1 (1) | 3 (3) |
| HL | CUYG_tetraloop | 20 (44) | 20 (43) | 0 (0) | 0 (0) |
| HL | AA_5SrRNA | 19 (19) | 18 (18) | 0 (0) | 0 (0) |
| HL | ANYA_tetraloop | 18 (39) | 16 (35) | 0 (0) | 1 (1) |
| IL | J4a/4b | 16 (37) | 16 (35) | 0 (0) | 1 (1) |
| J3 | J3_typeB | 16 (16) | 14 (14) | 2 (2) | 0 (0) |
| J3 | J3_hammerhead | 16 (27) | 16 (26) | 0 (0) | 1 (1) |
| HL | L8_RNaseP_bact_a | 16 (29) | 14 (26) | 0 (0) | 0 (1) |
| BS | LoopE_a.rev | 16 (21) | 13 (17) | 0 (0) | 1 (1) |
| HL | U_turn | 15 (24) | 14 (23) | 0 (0) | 0 (1) |
| HL | GANC_tetraloop | 15 (28) | 14 (26) | 0 (0) | 2 (2) |
| BS | CRC_binding.rev | 15 (18) | 8 (11) | 6 (6) | 0 (0) |
| BS | CsrA_motif | 15 (18) | 13 (16) | 0 (0) | 0 (1) |
| J3 | J3_TPP | 14 (19) | 12 (17) | 0 (0) | 0 (1) |
| HL | YGNN_tetraloop | 13 (27) | 12 (26) | 0 (0) | 0 (0) |
| IL | GUA_GG_RRE | 10 (13) | 10 (12) | 0 (0) | 0 (0) |
| HL | CsrA_binding | 10 (29) | 7 (14) | 0 (1) | 0 (0) |
| HL | GAGUA_pentaloop | 10 (22) | 9 (21) | 0 (0) | 0 (0) |
| IL | S_domain | 10 (21) | 10 (20) | 0 (0) | 0 (0) |
| IL | Loop_E | 9 (11) | 9 (10) | 0 (0) | 1 (2) |
| IL | Twisted_up | 9 (31) | 9 (30) | 0 (0) | 0 (0) |
| J3 | J3_purine | 7 (11) | 7 (11) | 0 (0) | 0 (0) |
| IL | GAAA_Tetraloop_receptor | 7 (17) | 7 (17) | 0 (0) | 0 (0) |
| HL | U6_loop | 7 (18) | 7 (18) | 0 (0) | 0 (1) |
| IL | J4/5_IL | 6 (13) | 6 (13) | 0 (1) | 0 (0) |
| HL | Sarcin_ricin_loop | 5 (7) | 5 (7) | 0 (0) | 1 (1) |
| IL | G_bulge_Das | 5 (11) | 5 (11) | 0 (0) | 0 (1) |
| IL | Docking_elbow_IL | 0 (1) | 0 (1) | 0 (0) | 0 (0) |

Rfam SEEDS
4,178 alignments

| | # Motifinstances with cov support (all) | # Rfam families with motif with cov support (all) | SSU eukarya with support (all) | LSU eukarya with support (all) |
|---|---|---|---|---|
| 3D Motifs all | 1460 (2124) | 591 (822) | 45 (51) | 62 (74) |
| Nested helices | 3877 (15395) | 1721 (4168) | 66 (89) | 102 (152) |
| PK + higher order | 504 (532) | 246 (246) | 1 (2) | 2 (4) |

Rfam CONTROLS
4,178 shuffled alignments

| | | | | |
|---|---|---|---|---|
| 3D Motifs all | 121 (290) | 106 (208) | 0 (0) | 0 (1) |
| Nested helices | 733 (14146) | 676 (4149) | 0 (34) | 0 (75) |
| PK + higher order | 175 (188) | 102 (102) | 0 (0) | 0 (0) |

Results of running CaCoFold-R3D for all seed alignments in the database of structured RNAs Rfam. Results show RNA 3D motif occurrences with covariation support where at least one of the closing helices has at least one covarying base pair, and in parenthesis the total set of predictions. For the other structural elements (nested helices, pseudoknots and other higher order base pair interactions), having covariation support means that the element includes at least one significantly covarying pair of residues. In parenthesis we show the total number of predictions. The control shuffled alignments were obtained by randomizing the residues within each alignment column independently from each other. In these control alignments, covariation between columns is scrambled, but the base composition per column (thus the possible 3D motif identity) remains mostly intact. Results of running CaCoFold-R3D on the Rfam seed alignments were obtained using the command line `R-scape -s –cacofold –r3d Rfam.seed`. The control experiments were obtained using the command line `R-scape -s –cacofold –r3d –vshuffle Rfam.seed`.

# Reporting Summary

## Statistics

For all statistical analyses, confirm that the following items are present in the figure legend, table legend, main text, or Methods section.

| n/a | Confirmed | |
|---|---|---|
| ☐ | ☒ | The exact sample size (*n*) for each experimental group/condition, given as a discrete number and unit of measurement |
| ☒ | ☐ | A statement on whether measurements were taken from distinct samples or whether the same sample was measured repeatedly |
| ☐ | ☒ | The statistical test(s) used AND whether they are one- or two-sided *Only common tests should be described solely by name; describe more complex techniques in the Methods section.* |
| ☐ | ☒ | A description of all covariates tested |
| ☐ | ☒ | A description of any assumptions or corrections, such as tests of normality and adjustment for multiple comparisons |
| ☒ | ☐ | A full description of the statistical parameters including central tendency (e.g. means) or other basic estimates (e.g. regression coefficient) AND variation (e.g. standard deviation) or associated estimates of uncertainty (e.g. confidence intervals) |
| ☐ | ☒ | For null hypothesis testing, the test statistic (e.g. *F*, *t*, *r*) with confidence intervals, effect sizes, degrees of freedom and *P* value noted *Give P values as exact values whenever suitable.* |
| ☒ | ☐ | For Bayesian analysis, information on the choice of priors and Markov chain Monte Carlo settings |
| ☒ | ☐ | For hierarchical and complex designs, identification of the appropriate level for tests and full reporting of outcomes |
| ☒ | ☐ | Estimates of effect sizes (e.g. Cohen's *d*, Pearson's *r*), indicating how they were calculated |

*Our web collection on statistics for biologists contains articles on many of the points above.*

## Software and code

Policy information about availability of computer code

| Data collection | Rfam 15.0 |
|---|---|
| Data analysis | R-scape v 2.5.7<br>Tornado v0.6 |

For manuscripts utilizing custom algorithms or software that are central to the research but not yet described in published literature, software must be made available to editors and reviewers. We strongly encourage code deposition in a community repository (e.g. GitHub). See the Nature Portfolio guidelines for submitting code & software for further information.

## Data

Policy information about availability of data

All manuscripts must include a data availability statement. This statement should provide the following information, where applicable:
- Accession codes, unique identifiers, or web links for publicly available datasets
- A description of any restrictions on data availability
- For clinical datasets or third party data, please ensure that the statement adheres to our policy

All data and results are available in the Supplemental Material

# Research involving human participants, their data, or biological material

Policy information about studies with human participants or human data. See also policy information about sex, gender (identity/presentation), and sexual orientation and race, ethnicity and racism.

| | |
|---|---|
| Reporting on sex and gender | n/a |
| Reporting on race, ethnicity, or other socially relevant groupings | n/a |
| Population characteristics | n/a |
| Recruitment | n/a |
| Ethics oversight | n/a |

Note that full information on the approval of the study protocol must also be provided in the manuscript.

# Field-specific reporting

Please select the one below that is the best fit for your research. If you are not sure, read the appropriate sections before making your selection.

☒ Life sciences  ☐ Behavioural & social sciences  ☐ Ecological, evolutionary & environmental sciences

For a reference copy of the document with all sections, see nature.com/documents/nr-reporting-summary-flat.pdf

# Life sciences study design

All studies must disclose on these points even when the disclosure is negative.

| | |
|---|---|
| Sample size | All 4,178 alignments in the Rfam database of structural RNA families were analyzed |
| Data exclusions | There was no data exclusion |
| Replication | The RNA structure+3D motif predictions are deterministic and do not change by replication |
| Randomization | Randomized alignments are generated internally by the code to provide a null model of scores in the absence of structure. The null model scores are used to calculate p-values and E-values. |
| Blinding | All RNA sequences testes had all existing structural information removed before testing. |

# Reporting for specific materials, systems and methods

We require information from authors about some types of materials, experimental systems and methods used in many studies. Here, indicate whether each material, system or method listed is relevant to your study. If you are not sure if a list item applies to your research, read the appropriate section before selecting a response.

## Materials & experimental systems

| n/a | Involved in the study |
|---|---|
| ☒ ☐ | Antibodies |
| ☒ ☐ | Eukaryotic cell lines |
| ☒ ☐ | Palaeontology and archaeology |
| ☒ ☐ | Animals and other organisms |
| ☒ ☐ | Clinical data |
| ☒ ☐ | Dual use research of concern |
| ☒ ☐ | Plants |

## Methods

| n/a | Involved in the study |
|---|---|
| ☒ ☐ | ChIP-seq |
| ☒ ☐ | Flow cytometry |
| ☒ ☐ | MRI-based neuroimaging |

