## [Peer Review File · Nature Methods]

All-at-once RNA folding with 3D motif prediction framed by evolutionary information

Corresponding Author: Dr Elena Rivas

Version 0:

Decision Letter:

11th Mar 2025

Dear Elena,

Your Article, "All-at-once RNA folding with 3D motif prediction framed by evolutionary information", has now been seen by 3 reviewers. As you will see from their comments below, although the reviewers find your work of considerable potential interest, they have raised a number of concerns. We are interested in the possibility of publishing your paper in Nature Methods, but would like to consider your response to these concerns before we reach a final decision on publication.

We therefore invite you to revise your manuscript to address these concerns. We recommend that in addition to addressing all the technical concerns, you make a stronger case for the utility and significance of the approach, specifically highlighting how the method stands out from others in terms of enabling discovery. Please also ensure that the code is user-friendly and can be tested.

Link Redacted

We hope to receive your revised paper within 6-8 weeks. If you cannot send it within this time, please let us know. In this event, we will still be happy to reconsider your paper at a later date so long as nothing similar has been accepted for publication at Nature Methods or published elsewhere.

OPEN SCIENCE REQUIREMENTS

REPORTING SUMMARY AND EDITORIAL POLICY CHECKLISTS

IMAGE INTEGRITY

EXTENDED DATA FIGURES

DATA AVAILABILITY

All novel DNA and RNA sequencing data, protein sequences, genetic polymorphisms, linked genotype and phenotype data, gene expression data, macromolecular structures, and proteomics data must be deposited in a publicly accessible database, and accession codes and associated hyperlinks must be provided in the "Data Availability" section.

Please include a "Data availability" subsection in the Online Methods. This section should inform readers about the availability of the data used to support the conclusions of your study, including accession codes to public repositories, references to source data that may be published alongside the paper, unique identifiers such as URLs to data repository entries, or data set DOIs, and any other statement about data availability. At a minimum, you should include the following statement: "The data that support the findings of this study are available from the corresponding author upon request", describing which data is available upon request and mentioning any restrictions on availability. If DOIs are provided, please include these in the Reference list

(authors, title, publisher (repository name), identifier, year). For more guidance on how to write this section please see: <http://www.nature.com/authors/policies/data/data-availability-statements-data-citations.pdf>

CODE AVAILABILITY

Please include a "Code Availability" subsection in the Online Methods which details how your custom code is made available. Only in rare cases (where code is not central to the main conclusions of the paper) is the statement "available upon request" allowed (and reasons should be specified).

MATERIALS AVAILABILITY

SUPPLEMENTARY PROTOCOL

To help facilitate reproducibility and uptake of your method, we ask you to prepare a step-by-step Supplementary Protocol for the method described in this paper. We [encourage authors to share their step-by-step experimental protocols](https://www.nature.com/nature-research/editorial-policies/reporting-standards#protocols) on a protocol sharing platform of their choice and report the protocol DOI in the reference list. Nature Portfolio's protocols.io is a free-to-use and open resource for protocols; protocols deposited onto protocols.io are citable and can be linked from the published article. More details can found at [protocols.io](https://www.protocols.io/help/publish-articles).

ORCID

Sincerely,
Arunima

Arunima Singh, Ph.D.
Senior Editor
Nature Methods

Reviewers' Comments:

Reviewer #1 (Remarks to the Author):

The authors present an updated version of R-scape/CaCoFold which now includes CaCoFold-R3D, an algorithm for predicting motifs found in alignments based upon covariation. This method is novel in that it predicts motifs while predicting a secondary structure using covariation and can predict a very large number of motifs at once. Other methods to predict RNA motifs cannot do both at once. This has the potential to be extremely useful to researchers interested in studying RNA structure. The overall presentation is clear and the conclusions supported by the data presented.

Major Comments

- The authors mention that this method can handle 'most' types of motifs in hairpins, internal loops and multiway loops. Could

they clarify which motifs cannot be modeled? Naively, I would expect this method will work for any motif in an IL or HL that are composed of internal interactions, while there may be some higher order junctions that may not yet be modeled. Could the authors clarify this point?

- How did the authors build the selected motifs? Did they base them on 3D structures, or alignments? Could they provide a little more detail on the process to create these models? Similarly, it would be useful if the authors provided a guide to how to model new motifs. While this maybe isn't a good fit for the manuscript it would be nice the user guide. I apologize if I missed this section of the guide, I only skimmed it.

- Can the building of motifs be automated? This would be useful and providing guidance on how to do this would be very helpful to users. Two examples would be to 1) use one of the motif libraries they cite as a source for all known motifs or 2) given an alignment which likely contains a motif, would it be possible to build a model to find more examples? Both seem like natural applications, but it is unclear how to do so from the manuscript.

Minor Comments

- How does this method work with junctions which contain internal canonical basepairs? For example, in the ribosomal small subunit the 3-way junction between h41, h42, h30 and h29 contains an embedded cWW basepair. This interaction shows no covariation, and very little variation and is needed for the overall motif structure. Does this loop get modeled as 4-way junction or split into two loops?

- The authors mention this method was trained on a dataset (line 113). Could they clarify the details of the dataset?

- How does this method handle large insertions within loops, e.g. insertions within K-turn. Does the method still identify these as K-turns?

- Could the authors clarify the results in table S3? Specifically what command is used to run R-scape?

Reviewer #1 (Remarks on code availability):

I was not able to run the code, I have an unusual setup and did not have enough time to update it to properly install R-scape.

Reviewer #2 (Remarks to the Author):

This paper proposes CaCoFold-R3D, a novel probabilistic grammar model for simultaneous prediction of RNA secondary structure and tertiary motifs. This model utilizes covariation information present in alignments to predict both secondary structure and tertiary motifs of RNA. The paper claims that CaCoFold-R3D can predict over 50 known RNA motifs almost exhaustively and can detect motifs appearing in any loop region. Furthermore, it integrates the prediction of RNA tertiary motifs and secondary structure using probabilistic modeling. CaCoFold-R3D can predict RNA structures considering sequence variations within a specific structural RNA family by using alignments as input. It is computationally efficient and easily customizable for novel motif discovery.

The strengths of this paper include:

- Ability to predict a wide range of RNA motifs.
- Integration of tertiary motif and secondary structure prediction through probabilistic modeling.
- Consideration of sequence variations in motif prediction by using alignments.
- Computational efficiency.
- Potential to facilitate novel motif discovery.

While the paper states that CaCoFold-R3D can handle pseudoknots, the specific method for doing so is not detailed. A more thorough explanation is needed, particularly regarding whether there are cases where covariation information and 3D motif information conflict in the presence of pseudoknots, and how such cases are addressed.

MC-Fold, RNAwolf, and RNA-MoIP all utilize MC-Sym for handling 3D motifs. A more detailed discussion comparing and contrasting these methods with CaCoFold-R3D is requested. In particular, the advantages and disadvantages of CaCoFold-R3D's approach of modeling 3D motifs directly using its own probabilistic grammar model, rather than relying on MC-Sym, should be clearly presented.

Reviewer #3 (Remarks to the Author):

The identification of RNA 3D motifs can identify functional elements in RNAs and guide the assembly of RNA tertiary structures. This manuscript presents CaCoFold-R3D, a method to jointly predict RNA secondary structures and 3D motifs with evolutionary support. CaCoFold-R3D constructs 51 distinct RNA 3D motifs derived from structured RNAs and develops a Stochastic Context-Free Grammar (SCFG), termed RBGJ3J4-R3D, based on these motifs and their variants. The model successfully predicts RNA 3D motifs within known RNA structures and has the potential to discover new one. CaCoFold-R3D thus has a great potential for studying RNA higher-order structures. However, several important issues need to be addressed.

Major comments:

1. The manuscript often lacks methodology details. For example, it provides limited information regarding the parameterization of RBGJ3J4-R3D and HMM data (e.g., "These parameters could be trained by maximum likelihood from datasets for RNA structures annotated with the 3D motifs."). Please offer a more detailed methodology to prevent potential data leakage issues. It makes the judge of the work challenging.
2. There are no details about the training dataset utilized by CaCoFold-R3D, and it appears there is no deduplication between the training and test sets, raising concerns about the reliability of the reported performance.

3. The manuscript should clarify the selection criteria and validation process for the 51 motif descriptors. Specifically: (1) What methodology was employed to establish these descriptors? Is there evidence supporting their reliability and quality? (2) How was their representativeness evaluated?
4. The article conducts an ablation study on the input of covariation information for R3D-prototype across 9 Rfam families and 2 RNA 3D motif types. However, it lacks details on how this study was performed (e.g., how it can be used without covariation constraints, lines 218-219). Additionally, it's unclear why the same ablation study was not conducted on CaCoFold-R3D or the entire dataset, as only shuffled alignments are used as controls.
5. The section "R3D-prototype: The importance of framing 3D motifs by evolutionary information" utilizes a demo model constructed with RBG-R3D grammar to validate that integrating evolutionary information enhances sensitivity in recognizing GNRA and K-turn motifs. However, these results may not directly generalize to the RBGJ3J4-R3D grammar. Can a similar analysis be performed using the RBGJ3J4-R3D grammar?
6. The manuscript insufficiently describes how the current approach can be generalized to new RNA 3D motifs or aid in novel motif discovery. Although the section "A new 3-way junction motif with high representation" provides an example, the methodological details remain vague.
7. CaCoFold-R3D utilizes evolutionary information for RNA secondary structure and 3D motif prediction. Notably, RNA multiple sequence alignment (MSA) computation can be technically challenging. This study focuses on evolutionarily informative sequences from Rfam. Could the authors provide additional analyses regarding MSA quality or single sequences to demonstrate the method's effectiveness in broader scenarios?
8. While validating with Rfam data, the manuscript displays only a few cases. Can you include statistical analyses that encompass global sensitivity measurements and motif-specific performance metrics (sensitivity and FDR)? It appears that the quality of descriptors may significantly influence results. For instance, the consensus sequences of J3_typeA and J3_typeB consist entirely of N, which may lead to elevated FDR levels.
9. The potential applications of RNA 3D motif prediction within the broader context of RNA structure and functional studies are not entirely clear. Additionally, there are no benchmark studies assessing the performance of CaCoFold-R3D, making it challenging to determine how well it performs or whether it outperforms other tools. For example, a quantitative comparison with existing methods such as RMDetect, JAR3D, RMfam, and BayesPairing2 would be necessary to further validate the advantages of the proposed method.

Minor comments:

- a) In Figure 2, there are inaccuracies and omissions in the degenerate base abbreviations: the notation "W = A/G" appears twice, while the abbreviation for "R" is missing.
- b) In the legend of Figure 2, the description "in purple positions of the motif involved in non-Watson-Crick base pairing, in green motif positions not paired" seems incorrectly labeled.
- c) On lines 116-117, "HL, BL, HL, J3, J4, and BS" should be corrected to "HL, BL, IL, J3, J4, and BS."
- d) On line 121, "could be train" should be revised to "could be trained."
- e) In Figure 3d S2 (5'→3') for J3_hammerhead, "CUGAGUA" should be corrected to "CUGAUGA."
- f) The manuscript states that "The total number of motifs implemented by CaCoFold-R3D after considering all motif variants is 96." However, our calculations indicate that the variant count is actually 97.
- g) On line 327, "RoseTTAFold" should be corrected to "RFAA." RoseTTAFold is specifically used for protein structure prediction.
- h) In the legend of Figure S3, "CaCoFold internally constructs SCFGs for a total of 96 motif variants" should be revised to "CaCoFold-R3D constructs SCFGs."

Version 1:

Decision Letter:

Our ref: NMETH-A59153A

29th Apr 2025

Dear Elena,

Thank you for submitting your revised manuscript "All-at-once RNA folding with 3D motif prediction framed by evolutionary information" (NMETH-A59153A). It has now been seen by the original referees and their comments are below. The reviewers find that the paper has improved in revision, and therefore we'll be happy in principle to publish it in Nature Methods, pending minor revisions to satisfy the referees' final requests and to comply with our editorial and formatting guidelines.

TRANSPARENT PEER REVIEW

ORCID

Sincerely,
Arunima

Arunima Singh, Ph.D.
Senior Editor
Nature Methods

Reviewer #1 (Remarks to the Author):

The authors have addressed my major concerns. Specifically, I feel that it is possible for someone with a new motif to be able to produce a model which R-scape can run. It may require some work for a novice user but it should be possible. The revision to include where the motifs came from is highly appreciated. Information on the training/test set gives more confidence in the method as well. Additionally, I can run the software and get results with expected motif predictions.

Reviewer #1 (Remarks on code availability):

I have been able to run the program but did not evaluate the code. I ran R-scape on a few families and manually inspected the results, which were as I would expect.

Reviewer #2 (Remarks to the Author):

Thank you for your appropriate responses to my comments.

Reviewer #3 (Remarks to the Author):

The revisions have addressed my concerns, with the presentation now demonstrating substantial improvements through enriched technical specifications and enhanced methodological clarity.

Version 2:

Decision Letter:

7th Aug 2025

Dear Elena,

I am pleased to inform you that your Article, "All-at-once RNA folding with 3D motif prediction framed by evolutionary information", has now been accepted for publication in Nature Methods. The received and accepted dates will be December 17, 2024 and August 7, 2025. This note is intended to let you know what to expect from us over the next month or so, and to let you know where to address any further questions.

Over the next few weeks, your paper will be copyedited to ensure that it conforms to Nature Methods style. Once your paper is typeset, you will receive an email with a link to choose the appropriate publishing options for your paper and our Author Services team will be in touch regarding any additional information that may be required. It is extremely important that you let us know now whether you will be difficult to contact over the next month. If this is the case, we ask that you send us the contact information (email, phone and fax) of someone who will be able to check the proofs and deal with any last-minute problems.

After the grant of rights is completed, you will receive a link to your electronic proof via email with a request to make any

corrections within 48 hours. If, when you receive your proof, you cannot meet this deadline, please inform us at rjsproduction@springernature.com immediately.

Authors may need to take specific actions to achieve compliance with funder and institutional open access mandates.

If your research is supported by a funder that requires immediate open access (e.g. according to [Plan S principles](https://www.springernature.com/gp/open-science/plan-s-compliance) or the [NIH public access policy](https://www.springernature.com/gp/open-science/us-federal-agency-compliance)) then you should select the gold OA route, and we will direct you to the compliant route where possible. Because authors warrant under our subscription licensing terms that they haven't committed to licensing any version of their article under a licence inconsistent with the terms of our agreement – including the applicable embargo period – publication under the subscription model isn't suitable for authors whose funders require no embargo.

If you are active on Twitter/X or Bluesky, please e-mail me your and your coauthors' handles so that we may tag you when the paper is published.

Best regards,
Arunima

Arunima Singh, Ph.D.
Senior Editor
Nature Methods

** Visit the Springer Nature Editorial and Publishing website at http://editorial-jobs.springernature.com?utm_source=ejP_NMeth_email&utm_medium=ejP_NMeth_email&utm_campaign=ejp_Nmeth www.springernature.com/editorial-and-publishing-jobs for more information about our career opportunities. If you have any questions please click [here](mailto:editorial.publishing.jobs@springernature.com). **

April 7, 2025

Arunima Singh, Ph.D.
Senior Editor
Nature Methods

Dear Dr. Singh,

Thank you for the reviews of our manuscript “All-at-once RNA folding with 3D motif prediction framed by evolutionary information” (NMETH-A59153), and your willingness to consider a revised version. We appreciate the positive comments provided by the reviewers, as well as their criticisms. We have revised the manuscript to address all points raised by yourself and the reviewers, which we have addressed as follows:

Reviewer #1

This method is novel in that it predicts motifs while predicting a secondary structure using covariation and can predict a very large number of motifs at once. Other methods to predict RNA motifs cannot do both at once. This has the potential to be extremely useful to researchers interested in studying RNA structure. The overall presentation is clear and the conclusions supported by the data presented.

Thank you for the positive summary of our work.

The authors mention that this method can handle most types of motifs in hairpins, internal loops and multiway loops. Could they clarify which motifs cannot be modeled? Naively, I would expect this method will work for any motif in an IL or HL that are composed of internal interactions, while there may be some higher order junctions that may not yet be modeled. Could the authors clarify this point?

Sorry for the ambiguous wording. The reviewer is correct that R3D can model any motif in ILs or HLs, as well as in 3-way or 4-way junctions. It cannot model interactions between branches in higher-order junctions, but it can model sequence motifs occurring in any of those branches independently from each other. We have modified the sentence accordingly.

How did the authors build the selected motifs? Did they base them on 3D structures, or alignments? Could they provide a little more detail on the process to create these models?

Our goal for this paper has been to consider a list as comprehensible as possible of 3D motifs. While there are many databases that collect individual motifs, however there is not one established database of RNA 3D motif models used by the whole community. Thus, we searched extensively the literature and introduced as many motifs as we could find reliably described in those methods. The new Table 2 provides information about the sources used to identify and select the motifs.

The motif models are mostly based on 3D structures. Sometimes, different groups provided slightly different descriptions for a given motif, and in some cases we have added both. For instance, the current default descriptor includes “G-bulge” (Cruz & Westhof, Nat Meth, 2011) and “G-bulge_Das” (Das et al, Nat Meth, 2010). Importantly, it is easy for CaCoFold-R3D to make instant adjustments to motif’s descriptions as desired.

The section [R3D SCFG profiles of over fifty recurrent RNA 3D motifs] together with the new Table 2 now include more information about the selection criteria, sorry about the previous lack of details.

Similarly, it would be useful if the authors provided a guide to how to model new motifs. While this maybe isnt a good fit for the manuscript it would be nice the user guide. I apologize if I missed this section of the guide, I only skimmed it.

Starting from an alignment, we decompose a motif into helical and non helical interactions. Helical interactions are modeled by classic Watson-Crick pairing, while non-helical interactions are broken down into contiguous subsequences that appear in loops; e.g. just one contiguous sequence in one loop closing a helix corresponds to a hairpin loop motif, while three separate loops connecting three helices correspond to a 3-way junction. We have added this information in the section [R3D: Six architectures to describe 3D motifs in all types of RNA loops].

We have also updated the R-scape userguide with information about how to model 3D motifs and how to add that information to a descriptor file that CaCoFold-R3D can read and implement automatically. The R-scape userguide can be found in the supplemental materials: [supplemental_material/src/CaCoFold-R3D/R-scape_userguide.pdf].

Can the building of motifs be automated? This would be useful and providing guidance on how to do this would be very helpful to users. Two examples would be to 1) use one of the motif libraries they cite as a source for all known motifs or 2) given an alignment which likely contains a motif, would it be possible to build a model to find more examples? Both seem like natural applications, but it is unclear how to do so from the manuscript.

In the manuscript, we present a direct example of scenario 2) with the new 3-way junction presented in Figure 6. Incorporating this 3-way junction into the model is automatic based on the methodology presented in Figure 3.

From one alignment (RF02001), where we have identified a likely motif, we have added the motif to the descriptor file (as described in Figure 6d), and we have identified many more instances of the same motif. In fact, it appears that this new 3-way junction is a very frequent motif.

In general, a motif candidate can be incorporated into our modeling framework in an algorithmic fashion (see response to previous comment on motif decomposition into helical and non-helical subsequences). Then run CaCoFold-R3D with the new model added allows us to identify new instances across families.

As for case 1), we look forward to using motif libraries to build more and better descriptors for CaCoFold-R3D. There is still work to be done by the scientific community in order to be able to use all the individual motifs displayed in databases such as the FR3D Motif library or CaRNAval to directly build the R3D profile HMMs for a given motif. We are in conversations with researchers from both groups to be able to achieve this important goal.

How does this method work with junctions which contain internal canonical basepairs? For example, in the ribosomal small subunit the 3-way junction between h41, h42, h30 and h29 contains an embedded cWW basepair. This interaction shows no covariation, and very little variation and is needed for the overall motif structure. Does this loop get modeled as 4-way junction or split into two loops?

Given four helices that can form a 4-way junction, the method considers all possible structural combinations and will report the one with the highest probability. Amongst those options, one would be

breaking the 4-way junction into two 3-way junction connected by one lone cWW pair. That is a possibility that the model will explore as the RBGJ3J4 grammar exhaustively evaluates all structures, and it allows lone pairs. In the end, the model reports the structure with the highest probability.

The situation would be different if the lone cWW basepair would show significant covariation. In that case, the covarying basepair would be included automatically, and that would result in a split into two 3-way junctions.

CaCoFold-R3D reports the max probability structure using the efficient CYK algorithm, constrained by the presence of all covarying pairs reported by R-scape. We have expanded the Methods section [*The RBGJ3J4-R3D joint grammar uses the CYK folding algorithm*] to describe the folding algorithm with more detail.

The authors mention this method was trained on a dataset (line 113). Could they clarify the details of the dataset?

We have added information about the datasets of RNA sequences/structures used by CaCoFold-R3D to train the RBGJ3J4 model. Those are the same used by CaCoFold to train the RBG model (Rivas, PLOSCB 2020). The datasets had been previously introduced in TORNADO (Rivas et al. 2012). They include a large collection of many RNA sequences from many different structural RNA families.

The old section [*Training*] is now split into two different sections: [*Training of RBGJ3J4 probabilistic parameters*] and [*Parameterization of RBGJ3J4-R3D probabilistic parameters*], which incorporate many more details about training and parameterization. Also see response to questions 1. and 2. from referee #3.

How does this method handle large insertions within loops, e.g. insertions within K-turn. Does the method still identify these as K-turns?

The method handles insertion in loops in two different ways. (1) Positions in the alignment that have more than 75% gaps (by default) are not analyzed. This effectively removes highly variable regions from analysis. (2) The profile HMMs describing each of the RNA 3D motifs allow for insertions and deletions between any two positions in a given consensus. As the number of insertions increases, the certainty in the occurrence of the motif will diminish, that would lower the probability of the motif, and eventually, it will be discarded in favor of just being modeled as a plain loop.

We have modified the section [*sequence profile HMM parameterization*] to stress these points.

Could the authors clarify the results in table S3? Specifically what command is used to run R-scape?

We understand that the referee is discussing Table S1 (the only supplemental table). We have added a line in the caption of Table S1 describing the command line used which just requires using the option `--r3d`.

In addition the supplemental material includes the full command lines used to produced all the results presented in this work, in the file [*supplemental_material/results/CaCoFold-R3D/00README*].

(Remarks on code availability): I was not able to run the code, I have an unusual setup and did not have enough time to update it to properly install R-scape.

We are sorry that the reviewer did not have time to install the code. We understand that it was not due to any intrinsic difficulty with the implementation.

Please see comment on the response to the Editor as to how we have taken additional steps to guide with a quick installation.

Reviewer #2

The strengths of this paper include:

- Ability to predict a wide range of RNA motifs.
- Integration of tertiary motif and secondary structure prediction through probabilistic modeling.
- Consideration of sequence variations in motif prediction by using alignments.
- Computational efficiency.
- Potential to facilitate novel motif discovery.

Thank you for a nice summary of the strengths of our work.

We could add to this list the fact that CaCoFold-R3D uses the evolutionary signal found in the alignment to inform the prediction both of canonical and 3D motifs which significantly improves performance, and provides statistical confidence on the predictions depending on the amount of covariation observed in the helices bounding the motif.

While the paper states that CaCoFold-R3D can handle pseudoknots, the specific method for doing so is not detailed. A more thorough explanation is needed, particularly regarding whether there are cases where covariation information and 3D motif information conflict in the presence of pseudoknots, and how such cases are addressed.

Sorry for the omission. The multilayered folding aspect of the CaCoFold-R3D folding approach that allows for the incorporation of pseudoknots was first introduced with CaCoFold (Rivas, PLOSCB, 2020). The new contributions of this work with the RBGJ3J4-R3D grammars occur all in the first layer. This first layer produces the main secondary structure along with the 3D motifs. Thus, we have concentrated most of our efforts describing the RBGJ3J4-R3D method and algorithms associated to it.

The incorporation of pseudoknots in CaCoFold-R3D occurs in the subsequent higher layers. Pseudoknots are helices of WC base pairs which preserve covariation when conserved just as the rest of helices in the main secondary structure. In the presence of non-nested covarying pairs that cannot be explained by the first layer, higher folding layers are added until all covarying pairs are taken into account. These higher layers use a simpler grammar that completes pseudoknotted helices when possible.

These higher layers may, in some cases, introduce conflicts with the first layer. We do not try to reconcile them. They may be conflicts, or instead they could indicate alternative structures. After all, these higher layers are there because there is still covariation signal not yet incorporated into a whole structure. We leave them for the user to decide.

This approach has proven quite successful identifying pseudoknots and integrating them with the detection of 3D motifs in the first layer. For instance, the Glutamine riboswitch in Figure 5 shows a motif found in a 3-way junction as well as a pseudoknot compatible with it (both confirmed by the crystal structure).

In the [Methods], we have added a section entitled [The CaCoFold-R3D multilayered folding and 3D motif

prediction method] describing these aspects of the algorithm.

MC-Fold, RNAwolf, and RNA-MoIP all utilize MC-Sym for handling 3D motifs. A more detailed discussion comparing and contrasting these methods with CaCoFold-R3D is requested. In particular, the advantages and disadvantages of CaCoFold-R3D's approach of modeling 3D motifs directly using its own probabilistic grammar model, rather than relying on MC-Sym, should be clearly presented.

MC-Sym infers atom positions, not 3D motifs explicitly, thus it is not directly comparable to CaCoFold-R3D.

It is our understanding that the programs MC-Fold, RNAwolf, and RNA-MoIP do not rely on MC-Sym, rather the outputs of the programs MC-Fold or RNAwolf can be used as inputs to MC-Sym. And RNA-MoIP is a wrapper of MC-Fold followed by MC-Sym.

MC-Fold builds a secondary structure with a model that introduces special pseudoenergy scores for some small whole hairpin loops and internal loops. However, the output does not identify any of the well known 3D motifs per se, and for example, it would not be able to tell that it has identified a K-turn from any other 2x5 internal loop.

RNAwolf also introduces terms to predict some triplets. However, RNAwolf cannot tell whether those triplets belong or constitute any of the known 3D motifs. CaCoFold directly predicts 3D motifs, thus a direct comparison is not possible. As we discuss in the manuscript, CaCoFold-R3D does not directly model the non-WC pairs, but it models loop branches that are correlated due to the presence of non-Watson-Crick in any arrangement, amongst those branches interactions they could be triplets. For example, the K-turn described in Figure 3, in which the G residue forms base pairs with both the R and A residues, we describe it as one interaction between the “G” and the “RA” profiles.

Moreover, none of these methods use evolutionary data in covariation to anchor the structure. One of the main points of this work is to stress the importance of framing the detection of 3D motifs by the covariation signal in the adjacent canonical helices.

We have included a new section [*Comparison to other methods*] where we describe the similarities and differences of CaCoFold-R3D with other related methods.

Reviewer #3

The model successfully predicts RNA 3D motifs within known RNA structures and has the potential to discover new one. CaCoFold-R3D thus has a great potential for studying RNA higher-order structures.

Thank you for considering our work of considerable potential interest.

However, several important issues need to be addressed.

1. The manuscript often lacks methodology details. For example, it provides limited information regarding the parameterization of RBGJ3J4-R3D and HMM data (e.g., These parameters could be trained by maximum likelihood from datasets for RNA structures annotated with the 3D motifs.). Please offer a more detailed methodology to prevent potential data leakage issues. It makes the judge of the work challenging.

2. There are no details about the training dataset utilized by CaCoFold-R3D, and it appears there is no deduplication between the training and test sets, raising concerns about the reliability of the reported performance.

These two points are addressed together. The method has three different modeling levels, and each level has its own parameterization.

- (1) The RBGJ3J4 SCFG (Figure 2a) for straight nested secondary structure prediction,
- (2) The RBGJ3J4-R3D extension (Figure 2b) to include all sort of 3D RNA motifs into one overall model and prediction, and
- (3) The individual R3D profile HMMs for each motif (Figure 3).

We emphasize that our training dataset is for parameters relevant to secondary structure, not parameters specific to 3D motifs. No information about 3D motif presence is included in the dataset, which is the downstream task our models are targeting. The old section *[Training]* has been split into two section addressing points (1) and (2) respectively.

(1) *[Training of RBGJ3J4 probabilistic parameters]*

This section describes the training of the RBGJ3J4 grammar for secondary structure which is similarly to that of the RBG grammar in CaCoFold. Both use the same training set consisting of a large collections of full RNA sequences/structures gathered from many different sources and representing a wide range of different RNA structures. This dataset was originally introduced with the method TORNADO with the purpose of testing the generalization of different methods for RNA structure prediction. Details of the dataset are provided in the manuscript.

(2) *[Parameterization of RBGJ3J4-R3D probabilistic parameters]*

The new rules added by RBGJ3J4-R3D to include 3D motifs cannot be trained with the dataset used in (1) because that dataset includes information about WC base pairs, but not about 3D motifs. A large and diverse dataset of full RNA sequences annotated with secondary structures as well as with 3D motifs has not been fully developed yet.

To overcome this lack of a training set including both complete secondary structures and 3D motifs, we use the maximum entropy principle to provide a RBGJ3J4-R3D parameterization in a way that avoids overfitting to any particular motif. More details are given in this new section.

(3) *[Parameterization of R3D profile HMMs]*

This section has been updated to describe with examples the parameterization of the sequence profile HMMs given a consensus sequence in a 3D motif. The emissions of the HMM match the sequence consensus allowing mismatches, and the transition probabilities are set for the model to allow some variability in the size of the motif.

We show that our parametrization generalizes over different RNA structures in 3D motif detection as confirmed both on the R3D prototype at the sequence level as well as on CaCoFold-R3D at the alignment level.

3. The manuscript should clarify the selection criteria and validation process for the 51 motif descriptors. Specifically: (1) What methodology was employed to establish these descriptors? Is there evidence supporting their reliability and quality? (2) How was their representativeness evaluated?

Since CaCoFold-R3D can incorporate many motifs simultaneously, our criteria was to have a comprehensive list of 3D motifs observed in RNA structures. The reliability and representativeness is based on the fact that these are all motifs described in the literature by major researchers in the field. We list all references that were used in the distillation of a comprehensive list. We have also added Table 2 which provides references for the provenance of the motif descriptors used.

The section *[R3D SCFG profiles of over fifty recurrent RNA 3D motifs]* has been updated to include information about the selection criteria for the 56 motifs selected in the current implementation.

We also notice that, would any of the selected motifs be considered suboptimal and requiring editing or removal, the only change that would have to be done to the whole method would be in the one line in the descriptor that describes the targeted motif. Similarly, would there be an important omission in the current list, it would only require adding a line to the descriptor incorporating the missing motif.

(see also comments given to a similar issue by reviewer #1)

4. The article conducts an ablation study on the input of covariation information for R3D-prototype across 9 Rfam families and 2 RNA 3D motif types. However, it lacks details on how this study was performed (e.g., how it can be used without covariation constraints, lines 218-219). Additionally, it's unclear why the same ablation study was not conducted on CaCoFold-R3D or the entire dataset, as only shuffled alignments are used as controls.

The R3D-prototype works on sequences, not alignments, thus by default the R3D-prototype runs without covariation constraints. As an option, one can add external information of covarying base pairs.

We have improved the description of the R3D-prototype, and have edited the original (218-219) lines. We emphasize that the prototype is a toy instantiation of CaCoFold-R3D at the sequence level that demonstrates how covariational information (here given as an add-on) can vastly improve 3D motif prediction.

As for a similar ablation study with CaCoFold-R3D on the entire Rfam data, that has indirectly been done. By performing the whole analysis on the Rfam data (Table S1), we can separate performance in motifs supported by covariation versus performance in motifs not supported by covariation which effectively acts as an ablation test. The FDR using shuffled controls is estimates at 8.3% for motifs supported by covariation, while the FDR becomes 25.4% for motifs not supported by covariation. Both numbers are now reported in the manuscript. We believe this is the first method of its kind that reports false discovery rates of its predictions at the alignment level. The new version of the manuscript includes these results in the section *[Results on RFAM alignments]*.

5. The section "R3D-prototype: The importance of framing 3D motifs by evolutionary information" utilizes a demo model constructed with RBG-R3D grammar to validate that integrating evolutionary information enhances sensitivity in recognizing GNRA and K-turn motifs. However, these results may not directly generalize to the RBGJ3J4-R3D grammar. Can a similar analysis be performed using the RBGJ3J4-R3D grammar?

We don't think this is necessary. As the new version of the manuscript now describes in more detail, the differences between RBG and RBGJ3J4 reside in the treatment of multiloops, and all the rest of the model and parameters are identical. Since the R3D-prototype only implements a HL motif (GNRA) and a IL motif (K-turn) but not any multiloop motif, it seems unlikely that any difference will occur. That is further

confirmed by the fact that CaCoFold-R3D with the RBGJ3J4 grammar detects the motifs tested by the prototype (Table 1) with 100% accuracy. The section [Results on RFAM alignments] reports this fact more clearly.

6. The manuscript insufficiently describes how the current approach can be generalized to new RNA 3D motifs or aid in novel motif discovery. Although the section "A new 3-way junction motif with high representation" provides an example, the methodological details remain vague.

What we provide with CaCoFold-R3D is an easy-to-use tool that gives the freedom to try and test many different options in a painless way. As we demonstrate with the 3-way junction, a candidate novel motif can be easily incorporated into CaCoFold-R3D, which then scans Rfam alignments efficiently to determine if this candidate is a recurring predicted motif across families. This provides an effective screening tool that aids in determining whether a structural pattern is prevalent amongst RNA families and can be classified as a motif.

We have added a paragraph discussing how the fact that CaCoFold-R3D works in alignments facilitates the identification not just of particular instances of the motif, as it is the case for other methods, but of the consensus and variability observed in the hypothetical motif.

We have added a paragraph in the discussion making the case on how our method stands out in terms of enabling discovery.

We have added a section in the R-scape userguide on how to model new motifs that are observed in a given alignment and are suspected to generalize to other structures.

7. CaCoFold-R3D utilizes evolutionary information for RNA secondary structure and 3D motif prediction. Notably, RNA multiple sequence alignment (MSA) computation can be technically challenging. This study focuses on evolutionarily informative sequences from Rfam. Could the authors provide additional analyses regarding MSA quality or single sequences to demonstrate the method's effectiveness in broader scenarios?

The quality of the alignment as well as the procedure used to build it (structural with Infernal or non structural with nhmmer or other structure-agnostic methods) are very important issues. We have already discussed several of those alignment issues in great detail in our previous work "Thirteen dubious ways to detect conserved structural RNAs" (Gao *et al.*, 2022).

The main points of this study with respect to alignments are: (1) you can make reliable 3D motif predictions informed by covariation found in good structural alignments, and (2) even within a good structural alignment one needs to pay attention to the supporting covariation for each motif because there is an important difference in confidence whether the prediction is bound by covariation or not.

We have added a paragraph at the end of the section [Results on RFAM alignments] to stress these points and to discuss possible pitfalls in alignments.

8. While validating with Rfam data, the manuscript displays only a few cases. Can you include statistical analyses that encompass global sensitivity measurements and motif-specific performance metrics (sensitivity and FDR)? It appears that the quality of descriptors may significantly influence results. For instance, the consensus sequences of J3_typeA and J3_typeB consist entirely of N, which may lead to elevated FDR levels.

We use detection rates in shuffled sequences as a measure of the global FDR, as there isn't an established enumeration of 3D motifs present and absent per alignment across RNA families.

In Table S1, we provide global FDR measures for all motifs and all Rfam families. None of the 3D motifs tested shows particularly large numbers of false positives. In particular, the J3_typeA has 4 false positives and J3_typeB has 6 false positives, those numbers are comparable to those of other motifs. The false positives per motif can be found in the file `[/supplemental_material/results/CaCoFold-R3D/CaCoFold-R3D_Rfam_vshuffle/Rfam.rscape.FamsPerMotif.tbl]`.

Assessing global sensitivity on 3D module recognition is not feasible for the existing data. Many of the Rfam families lack crystal structures, and there is not a standard in the field for systematic sensitivity benchmarking. We address the benchmarking on sensitivity in more detail in the next response (9.).

9. The potential applications of RNA 3D motif prediction within the broader context of RNA structure and functional studies are not entirely clear. Additionally, there are no benchmark studies assessing the performance of CaCoFold-R3D, making it challenging to determine how well it performs or whether it outperforms other tools. For example, a quantitative comparison with existing methods such as RMDetect, JAR3D, RMfam, and BayesPairing2 would be necessary to further validate the advantages of the proposed method.

We have added in the discussion a section about the utility and significance of this method in the context of RNA structure and functional studies, which include being

(1) A tool for the development of deep learning methods of RNA 3D structure prediction. Alignments are fundamental to enhance the signal to train models such as AlphaFold. CaCoFold-R3D extracts important 3D motifs information from alignments that can strongly inform RNA 3D structure prediction. These evolutionary structural features can be used in loss functions at different stages of training.

(2) A tool in RNA therapeutics based on RNA interacting with small molecules. Our method introduces important knowledge about the conformation of RNA loops in the context of a whole RNA structure, and it could greatly improve the design of RNAs with particular therapeutic functions based on the configuration of their loops and their interactions with small molecules.

(3) A tool to construct a comprehensive database of representative structural elements present in RNA structures. This database could have a similar relevance to database of protein domain.

As for benchmarking, the lack of a robustly labeled dataset marking what 3D motifs are present vs. not for each alignment makes an otherwise easy quantitative comparison between CaCoFold-R3D and other methods difficult. The next best comparison is demonstrating CaCoFold-R3D can identify the same motifs in alignments catalogued in the literature, while also identifying known motifs other approaches cannot.

We have introduced a new Table (Table 2) which presents positives for most of the 56 tested motifs of families with crystal structure support, including the 6 motifs total implemented by the previous methods. In addition, in the text and Figure 5, we have compiled and highlighted many families with known motifs for which CaCoFold-R3D identifies correctly the motifs, often in situations where the structure includes more than one motif as in the case of Metazoan SRP or the T-box riboswitch in Figure 5. No other method is able to predict all the different motifs present in those two structures.

Outside of benchmarking on what other methods are able to detect, we believe we have shown that CaCoFold-3D has enough unique features combined together that validates its contribution to computational RNA structure prediction. The methods mentioned by the reviewer, RMDetect, JAR3D, RMfam, and BayesPairing2, which are worthy attempts to identify 3D motifs, have limitations that would make any comparison unlevelled at best. For instance, they all need to provide a secondary structure, CaCoFold-R3D does not. They all take care of a much more limiting number of 3D motifs, and they do so one at the time. In addition, R-scape-R3D results are obtained with just one family-independent parameterization of CaCoFold-R3D that generalizes to all families. This is unlike BayesPairing2, for which results vary a lot depending on which family has been used to train the model. We have added a new section to provide a more detailed comparison to other methods.

Minor comments: a) In Figure 2, there are inaccuracies and omissions in the degenerate base abbreviations: the notation "W = A/G" appears twice, while the abbreviation for "R" is missing.

Thanks for noticing the typo. Figure 2 has been corrected to show "R=A/G" and "W=A/U".

b) In the legend of Figure 2, the description "in purple positions of the motif involved in non-Watson-Crick base pairing, in green motif positions not paired" seems incorrectly labeled.

The legend of Figure 2 has been corrected by swapping the words "purple" with "green".

c) On lines 116-117, HL, BL, HL, J3, J4, and BS should be corrected to HL, BL, IL, J3, J4, and BS.

Typo corrected in updated version.

d) On line 121, could be train should be revised to could be trained.

Typo fixed.

e) In Figure 3d S2 (53) for J3_hammerhead, CUGAGUA should be corrected to CUGAUGA.

Thanks for noticing the typo. Figure 3 has been corrected.

f) The manuscript states that The total number of motifs implemented by CaCoFold-R3D after considering all motif variants is 96. However, our calculations indicate that the variant count is actually 97.

Thanks for your careful analysis of the manuscript. The number of motifs in the descriptor file is 56 which are split into:

$$HL(15) + BL(3) + IL(17) + J3(7) + J4(4) + BL(10) = 56$$

HL and BL motifs have only one variant, while IL have 2 variants, J3 have 3, J4 have 4 and BL just one. Thus, the total number of motifs including variants is

$$HL(15) + BL(3) + 2 * IL(17) + 3 * J3(7) + 4 * J4(4) + BL(10) = 99$$

Moreover, all three variants of the J3 motif J3_typeB are identical, which reduces the count by 2. And both variants of the IL motif J4/5-IL are also identical, which reduces the count by one more. Thus, the total

number of non-identical variants is

$$99 - 2 - 1 = 96.$$

The R3D code, given a descriptor file, calculates automatically all possible variants and eliminates those that are redundant. The total number of total non-redundant RNA motifs used is reported as part of the standard output for further inspection. The updated manuscript includes more details about motif variants.

g) On line 327, RoseTTaFold should be corrected to RFAA. RoseTTaFold is specifically used for protein structure prediction.

Reference modified to “RoseTTaFold All-Atom (RFAA)”.

h) In the legend of Figure S3, CaCoFold internally constructs SCFGs for a total of 96 motif variants should be revised to CaCoFold-R3D constructs SCFGs.

Corrected.

Editor

We recommend that in addition to addressing all the technical concerns, you make a stronger case for the utility and significance of the approach, specifically highlighting how the method stands out from others in terms of enabling discovery.

We have updated the [Discussion] section in order to make a stronger case for the significance of the approach.

The core qualities of our method ability to model almost any motif, successful prediction on known motifs, joint secondary and tertiary prediction, computational speed that distinguishes CaCoFold-R3D from related algorithms in the literature also makes it effective as a tool for novel motif discovery. These qualities allow fast and reliable screening for discovering new, prevalent structural patterns that characterize 3D structure across RNA families.

We see our method as having two important applications. One is to provide valuable data to enhance the performance of deep learning methods for RNA 3D structure prediction, which is heavily affected by the small amount of data we currently have. The other one, is to have an impact in therapeutics grounded in small molecules interacting with RNA drug targets. These therapeutic compounds (such as risdiplam for spinal muscular atrophy) rely on the presence of RNA loops forming favorable binding pockets. Our method, which contains powerful predictive information of key structural elements in RNA loops should be able to guide improved design of new such therapeutic compounds.

Please also ensure that the code is user-friendly and can be tested.

We are sorry to hear that reviewer #1 did not have time to install it. We believe the code is user friendly and can be easily implemented in any Unix or Mac system (including the arm architecture). Nevertheless, we have taken additional steps to further facilitate installation and testing CaCoFold-R3D to anyone reading this manuscript.

We can assure after many tests and the feedback of multiple users that the software should install smoothly with just three commands (./configure;make;make install) after the tarball has been extracted (tar -xvf

rscope_v2.5.7.tar.gz). Installing the whole software does not require the installation of any other additional tools for it to work at full capacity.

To make installation easier, the above information has been explicitly added in the supplemental material, file: *[supplemental_material/src/CaCoFold-R3D/00README]*.

Sincerely,

Elena Rivas, Ph.D.

June 12, 2025

Arunima Singh, Ph.D.
Senior Editor
Nature Methods

Dear Dr. Singh,

Thank you for the reviews of our manuscript “All-at-once RNA folding with 3D motif prediction framed by evolutionary information” (NMETH-A59153A). The reviewers’ comments are all positive and do not require to introduce any updates to the manuscript.

Reviewer #1

The authors have addressed my major concerns. Specifically, I feel that it is possible for someone with a new motif to be able to produce a model which R-scape can run. It may require some work for a novice user but it should be possible. The revision to include where the motifs came from is highly appreciated. Information on the training/test set gives more confidence in the method as well. Additionally, I can run the software and get results with expected motif predictions.

We thank the reviewer for the comments.

(Remarks on code availability): I have been able to run the program but did not evaluate the code. I ran R-scape on a few families and manually inspected the results, which were as I would expect.

We are glad the reviewer was able to install the code and check the results.

Reviewer #2

Thank you for your appropriate responses to my comments.

We are glad the responses were satisfactory.

Reviewer #3

The revisions have addressed my concerns, with the presentation now demonstrating substantial improvements through enriched technical specifications and enhanced methodological clarity.

We thank the reviewer for the comments.

Sincerely,

Elena Rivas, Ph.D.